# Communication-Efficient Decentralized Optimization via Double-Communication Symmetric ADMM

**Jinrui Huang**[1]**, Xueqin Wang**[1]**, Dong Liu**[1,3]**, Jingguo Lan**[1]**, Runxiong Wu**[2*]

[1] Department of Statistics and Finance, University of Science and Technology of China
[2] Department of Industrial & Systems Engineering, University of Wisconsin–Madison
[3] Department of Operations Research and Financial Engineering, Princeton University

## Abstract

This paper focuses on decentralized composite optimization over networks without a central coordinator. We propose a novel decentralized symmetric ADMM algorithm that incorporates multiple communication rounds within each iteration, derived from a new constraint formulation that enables information exchange beyond immediate neighbors. While increasing per-iteration communication, our approach significantly reduces the total number of iterations and overall communication cost. We further design optimal communication rules that minimize the number of rounds and variables transmitted per iteration. The proposed algorithms are shown to achieve linear convergence under standard and relatively weak assumptions (e.g., metric subregularity). Extensive experiments on regression and classification tasks validate the theoretical results and demonstrate superior performance compared to existing decentralized optimization methods.

## 1 Introduction

The increasing size and complexity of modern machine learning models, combined with the explosive growth of data from sources such as mobile devices, sensors, and edge computing platforms, has driven the demand for scalable and privacy-preserving optimization techniques. Among these, decentralized optimization has emerged as a powerful approach, particularly when centralized computation is impractical due to concerns of scalability, robustness, data privacy, communicational infeasibility and network connectivity.

Unlike centralized distributed optimization, which still depends on a central server to coordinate updates, decentralized optimization involves multiple agents collaboratively solving a global problem by performing local computations and exchanging information only with their neighbors. These agents operate over a connected network—typically modeled as a graph—without the need for a central coordinator. This architecture makes decentralized optimization especially attractive for applications in multiple fields like sensor networks and large-scale machine learning.

In decentralized optimization, a central challenge lies in reducing the time cost of both local computation and inter-node communication. While existing algorithms differ in their local update strategies, most follow a common structural pattern: each iteration is followed by a single round of communication. This convention has seldom been challenged, primarily due to the concern that introducing multiple communication rounds per iteration would increase the overall communicational cost. Prior attempts to incorporate fixed multiple communication rounds, such as those gradient-based methods in (Jakovetić et al., 2014; Berahas et al., 2019; Ye et al., 2023; Li et al., 2018), achieve this by multi-consensus—repeatedly mixing local variables through communication. However, these methods have not demonstrated practical reductions in the total number of communication rounds. As a result, the potential for achieving a net communication reduction through non-adaptive multi-communication algorithms remains largely unexplored.

---

*Corresponding Author: Runxiong Wu (`rwu246@wisc.edu`)

Although multi-consensus schemes involve more frequent averaging steps, they do not necessarily reduce the overall communication cost. A potential reason is that these repeated consensus/communication steps primarily accelerate agreement among local variables, but offer limited improvement to the quality of each iteration. This observation motivates the need for a more principled and dedicated framework for multi-round communication, rather than simply applying multi-consensus in iteration.

In this paper, we explore incorporating multiple communication rounds within a single iteration in an approach orthogonal to existing multi-consensus schemes inspired by this observation. Rather than directly applying multiple mixing steps, we develop our algorithms by introducing linear constraints tailored for ADMM, which naturally embed multi-round communication into each iteration. To further enhance performance, we adopt a Symmetric ADMM (He et al., 2016) framework to accelerate convergence. Although this design increases the per-iteration communication cost, it enables a significantly faster convergence, leading to a substantial reduction in the total number of iterations, computations, and overall communication required for convergence.

Our contributions are summarized as follows:

- We propose DS-ADMM, a novel Symmetric ADMM-based decentralized composite optimization framework that incorporates multiple communication rounds into each iteration, leading to more efficient decentralized training.
- We derive optimal communication rules in the proposed algorithms which successfully minimize the communication rounds and the amount of information transmitted per iteration.
- We provide rigorous theoretical guarantees for the proposed method, including convergence and convergence rate analysis under standard and strong convexity assumptions.
- We conduct extensive numerical experiments that validate our theoretical results and demonstrate the superior performance of our method compared to state-of-the-art algorithms in decentralized composite optimization by reducing both computational and communicational cost.

To the best of our knowledge, such a symmetric-ADMM construction with a fixed multi-round schedule and the associated theory has not been explored in prior decentralized optimization literature, and it successfully reduces the overall communication cost. Our results open a promising direction for decentralized optimization by revealing a new trade-off between the number of per-iteration communication rounds and overall convergence speed.

## 1.1 RELATED WORK

**Decentralized Optimization.** Decentralized optimization has gained significant attention in large-scale machine learning, particularly in scenarios where data is distributed across multiple agents or devices without a central coordinator. A common strategy in these methods is the use of stochastic or deterministic mixing matrices to perform local averaging of variables across neighboring nodes, facilitating global consensus through communication. Early methods such as Decentralized Gradient Descent (DGD) (Nedic & Ozdaglar, 2009; Jakovetić et al., 2012; Yuan et al., 2016) directly extend classical gradient-based algorithms to networked environments, laying the groundwork for later developments. However, DGD suffers from slow convergence and sensitivity to step size. To address these shortcomings, a line of gradient-tracking based algorithms has been developed, including EXTRA and PG-EXTRA (Shi et al., 2015a;b), NIDS (Li et al., 2019), SONATA (Sun et al., 2022), and (Qu & Li, 2018), which incorporate correction terms to estimate the average gradient across the network. Several of these algorithms, including (Shi et al., 2015b; Li et al., 2019; Sun et al., 2022) are designed for decentralized composite optimization problems with smooth-nonsmooth structure by embedding proximal gradient steps. And the unifying analysis in (Xu et al., 2021) shows that a broad family of methods—including (Shi et al., 2015b; Li et al., 2019; Qu & Li, 2018)—can be understood through a single theoretical framework and established linear convergence under strong convexity assumptions. In addition to improving per-iteration convergence rates, many recent methods incorporate acceleration techniques such as Nesterov acceleration, leading to further improvements in both computation and communication complexity including (Jakovetić et al., 2014; Li et al., 2018; Qu & Li, 2020; Ye et al., 2023). In parallel, a different family of methods focuses on dual-based formulations. These include (Uribe et al., 2020; Scaman et al., 2017; Lan et al., 2017)

and decentralized adaptations of the Alternating Direction Method of Multipliers (ADMM) (Wei & Ozdaglar, 2013; Shi et al., 2014; Chang et al., 2015; Aybat et al., 2018) which reformulate the problem with consensus constraints and alternate between primal and dual updates. A notable development in this line is (Wang et al., 2018), which construct constraints based on mixing matrix into an ADMM framework for decentralized composite optimization.

**ADMM and Symmetric ADMM.** The Alternating Direction Method of Multipliers (ADMM) is a powerful algorithmic framework for solving linearly constrained convex optimization problems with separable objective structures, and it does so without requiring smoothness assumptions. This characteristic makes ADMM particularly well-suited for composite optimization tasks involving non-smooth loss functions and regularizers. We refer readers to (Eckstein & Yao, 2015; Boyd et al., 2011; Glowinski, 2014) for comprehensive overviews of the classical ADMM framework, and to (Deng & Yin, 2016; He & Yuan, 2015; Han et al., 2018; Yuan et al., 2020) for detailed convergence analyses of ADMM and its various extensions. To improve convergence speed and numerical performance, Symmetric ADMM (S-ADMM) and its generalizations have been proposed in recent years (He et al., 2016; Bai et al., 2018). These methods modify the standard ADMM iteration by introducing a symmetric primal-dual update structure, typically involving an additional intermediate update of the dual variable. This symmetric design allows for more balanced update dynamics between the primal and dual variables and often leads to improved practical performance. Convergence analyses for S-ADMM and its extensions have been established in works such as (Bai et al., 2019; Gu et al., 2015).

**Multi-Communication in Decentralized Optimization.** Incorporating multiple communication rounds per iteration has been explored in decentralized optimization for different purposes, using either fixed or adaptive strategies. Early work by (Berahas et al., 2019) showed that fixed multi-communication inherits DGD's convergence issues and incur high communication costs, while adaptive strategies—where communication rounds increase periodically—can achieve exact convergence, albeit requiring tuning or prior knowledge. Later methods (Jakovetić et al., 2014; Ye et al., 2023; Li et al., 2018) adopted fixed multi-communication to attain optimal theoretical communication complexity. However, empirical results in (Ye et al., 2023) indicate that fixed multi-round schemes are unable to reduce total communication. This limitation motivates algorithmic designs that embed multi-communication within the problem structure, rather than treating it as an external enhancement.

## 2 PRELIMINARIES

### 2.1 PROBLEM SETUP

We consider a network of $n$ agents collaboratively solving the decentralized composite optimization problem

$$\min_{x \in \mathbb{R}^d} F(x) = \sum_{i=1}^n \big[ f_i(x) + g_i(x) \big], \tag{1}$$

where $f_i$ is a convex local loss function and $g_i$ is a convex local regularizer, both privately held by agent $i$. In settings involving a global regularizer $g$, we distribute it across the agents as $g_i = p_i g$ with nonnegative weights satisfying $\sum_{i=1}^n p_i = 1$. This formulation captures a broad range of decentralized machine learning problems. Typical examples of loss functions include least squares, quantile loss, Huber loss, and hinge loss, while common regularizers include the $\ell_1$-norm, $\ell_2$-norm, and elastic net. Moreover, we denote the proximal operator of function $f$ as

$$\text{Prox}_{\eta f}(y) = \arg\min_{x \in \mathbb{R}^m} \left( f(x) + \frac{1}{2\eta} \|x - y\|^2 \right) \tag{2}$$

Many commonly used loss and regularization functions admit closed-form proximal operators, which we leverage in the algorithmic development that follows.

### 2.2 GRAPH TOPOLOGY

We model the communication pattern among agents using an undirected and connected graph $G = (V, E)$, where $V = \{1, 1, \dots, n\}$ denotes the set of agents and an edge $(i, j) \in E$ represents

a direct bidirectional communication link between agents $i$ and $j$. The graph structure is encoded by a mixing matrix $W \in \mathbb{R}^{n \times n}$, where $W_{ij} \in [0, 1]$ specifies the communication weight assigned to agent $j$ by agent $i$. Throughout the paper, we impose the following standard assumptions on $W$:

**Assumption 1.** *The mixing matrix $W$ satisfies: (1) $W$ is symmetric; (2) $W$ is doubly stochastic, i.e., $W\mathbf{1} = \mathbf{1}$, where $\mathbf{1}$ denotes the all-ones vector; (3) $W_{ij} > 0$ for $i \neq j$ if and and only if $(i, j) \in E$, and $W_{ii} > 0$ for all $i \in V$.*

An example of such a matrix based on the Metropolis–Hastings weights (Hastings, 1970) is provided in Appendix A. Under Assumption 1, the graph and its mixing matrix enjoy several well-known properties that follow from the Perron–Frobenius theorem:

**Proposition 1.** *The following properties hold: (1) The eigenvalues of $W$ satisfy $1 = \lambda_1(W) > \lambda_2(W) \geq \cdots \geq \lambda_n(W) > -1$, and the spectral gap $\rho = 1 - \max\{|\lambda_2(W)|, |\lambda_n(W)|\}$ is strictly positive; (2) The null space of $I_n - W$ is given by $\mathrm{null}(I_n - W) = \mathrm{span}\{\mathbf{1}\}$, which characterizes the consensus subspace.*

## 3 METHOD

This section presents the design of our communication-efficient decentralized algorithm. We begin by reformulating the consensus constraint to enable multiple rounds of communication within each iteration. We then explain how proximal linearization facilitates decentralized updates, describe our communication scheme that minimizes per-iteration transmission, and finally introduce the full decentralized form of the proposed algorithm.

### 3.1 SYMMETRIC CONSENSUS CONSTRAINTS

A key requirement for applying symmetric ADMM (He et al., 2016) in decentralized optimization is to enforce agreement among agents' local variables. Unlike distributed ADMM (Boyd et al., 2011), where a central coordinator explicitly enforces consistency, the decentralized setting lacks a global authority and therefore requires a different mechanism for encoding consensus within linear constraints. Our approach is motivated by the spectral properties of the mixing matrix $W$, in particular the structure of the null space of $I_n - W$. Let $\widetilde{W} = W \otimes I_d$, and let $u = (u_1^\top, \ldots, u_n^\top)^\top \in \mathbb{R}^{nd}$ denote the stacked vector of local variables. Using the fact that $W$ is symmetric, doubly stochastic, and has a positive spectral gap, one can show:

**Proposition 2** (Symmetric consensus constraint). *The consensus condition $u_1 = u_2 = \cdots = u_n$ is equivalent to the existence of an auxiliary stacked variable $v = (v_1^\top, \ldots, v_n^\top)^\top \in \mathbb{R}^{nd}$ satisfying $u = \widetilde{W}v$ and $v = \widetilde{W}u$.*

This pair of constraints in the above Proposition 2 embeds consensus into a linear, fully decentralized structure. Let $f(u) = \sum_{i=1}^{n} f_i(u_i)$ and $g(v) = \sum_{i=1}^{n} g_i(v_i)$. Using the above constraints, problem equation 1 is equivalently reformulated as

$$\min_{u,v \in \mathbb{R}^{nd}} f(u) + g(v) \qquad \text{s.t.} \qquad Au - Bv = 0, \tag{3}$$

where $A = (\widetilde{W}, I_{nd})^\top$ and $B = (I_{nd}, \widetilde{W})^\top$. Note that the reformulated problem equation 3 is invariant under the exchange $u \leftrightarrow v$ and $(A, B) \leftrightarrow (B, A)$. This invariance makes the associated symmetric ADMM iteration balanced and self-adjoint: the two primal blocks enter the augmented Lagrangian in exactly the same way and therefore admit identical ADMM updates. We refer to this formulation as the *symmetric consensus constraint* to highlight this structural novelty.

**Remark 1.** *The ADMM variant proposed in (Wang et al., 2018) uses a related but asymmetric constraint formulation with $A = \left((I_{nd} - \widetilde{W})^{1/2}, I_{nd}\right)^\top$ and $B = (0, I_{nd})^\top$. This design enables a single communication round per iteration by absorbing part of the dual update. However, the inherent asymmetry prevents the use of symmetric ADMM, which requires a balanced primal–dual structure.*

### 3.2 GRAPH-AWARE PROXIMAL LINEARIZATION

Let $\lambda = (\lambda_1, \lambda_2)$ denote the Lagrange multipliers associated with the two equality constraints in equation 3, let $\beta > 0$ be the augmented Lagrangian penalty parameter, and let $r, s > 0$ be the

symmetric dual update steps. The generalized symmetric ADMM (see, e.g., He et al., 2016; Bai et al., 2018; Deng & Yin, 2016) applied to the reformulated problem equation 3 takes the form

$$
\begin{cases}
u^{(t+1)} = \arg \min_{u \in \mathbb{R}^{nd}} f(u) - \langle \lambda^{(t)}, Au \rangle + \dfrac{\beta}{2} \|Au - Bv^{(t)}\|^2 + \dfrac{1}{2} \|u - u^{(t)}\|_Q^2, \\[2mm]
\lambda^{(t+\frac{1}{2})} = \lambda^{(t)} - r\beta \big(Au^{(t+1)} - Bv^{(t)}\big), \\[2mm]
v^{(t+1)} = \arg \min_{v \in \mathbb{R}^{nd}} g(v) + \langle \lambda^{(t+\frac{1}{2})}, Bv \rangle + \dfrac{\beta}{2} \|Au^{(t+1)} - Bv\|^2 + \dfrac{1}{2} \|v - v^{(t)}\|_Q^2, \\[2mm]
\lambda^{(t+1)} = \lambda^{(t+\frac{1}{2})} - s\beta \big(Au^{(t+1)} - Bv^{(t+1)}\big).
\end{cases}
\tag{4}
$$

To efficiently linearize the subproblems and remove the quadratic coupling introduced by the terms $\|Au - Bv^{(t)}\|^2$ and $\|Au^{(t+1)} - Bv\|^2$, we introduce a graph-aware positive definite proximal matrix

$$
Q \; = \; \beta\big((1+\tau)I_{nd} - \widetilde{W}^\top \widetilde{W}\big), \qquad \tau > 0.
$$

This choice ensures that the augmented Lagrangian becomes separable across agents, enabling fully decentralized updates of both primal variables. Then, by expanding the symmetric ADMM updates in equation 4, the updates can be rewritten in the following decomposable form:

$$
\begin{cases}
u^{(t+1)} = \arg \min_{u \in \mathbb{R}^{nd}} f(u) - \Big\langle \widetilde{W}\lambda_1^{(t)} + \lambda_2^{(t)} + \beta\Big(2\widetilde{W}v^{(t)} + (1+\tau)u^{(t)} - \widetilde{W}^\top \widetilde{W}u^{(t)}\Big), u \Big\rangle \\[2mm]
\qquad\qquad + \beta\Big(1 + \dfrac{\tau}{2}\Big)\|u\|^2, \\[2mm]
\lambda_1^{(t+\frac{1}{2})} = \lambda_1^{(t)} - r\beta\big(\widetilde{W}u^{(t+1)} - v^{(t)}\big), \; \lambda_2^{(t+\frac{1}{2})} = \lambda_2^{(t)} - r\beta\big(u^{(t+1)} - \widetilde{W}v^{(t)}\big), \\[2mm]
v^{(t+1)} = \arg \min_{v \in \mathbb{R}^{nd}} g(v) - \Big\langle \beta\Big(2\widetilde{W}u^{(t+1)} + (1+\tau)v^{(t)} - \widetilde{W}^\top \widetilde{W}v^{(t)}\Big) - \lambda_1^{(t+\frac{1}{2})} - \widetilde{W}\lambda_2^{(t+\frac{1}{2})}, v \Big\rangle \\[2mm]
\qquad\qquad + \beta\Big(1 + \dfrac{\tau}{2}\Big)\|v\|^2, \\[2mm]
\lambda_1^{(t+1)} = \lambda_1^{(t+\frac{1}{2})} - s\beta\big(\widetilde{W}u^{(t+1)} - v^{(t+1)}\big), \; \lambda_2^{(t+1)} = \lambda_2^{(t+\frac{1}{2})} - s\beta\big(u^{(t+1)} - \widetilde{W}v^{(t+1)}\big).
\end{cases}
\tag{5}
$$

This graph-aware linearization eliminates quadratic coupling, restores decomposability, and yields update rules that are fully decentralized and structurally balanced across the two primal blocks. For a detailed derivation of the general update form, we refer the reader to Appendix B.

### 3.3 OPTIMAL DOUBLE-COMMUNICATION STRUCTURE

We now describe how to implement the decomposable symmetric ADMM updates in equation 5 in a fully decentralized manner. There are four blocks of variables $u, \lambda_1, \lambda_2, v \in \mathbb{R}^{nd}$. We write

$$
u = (u_1^\top, \ldots, u_n^\top)^\top, v = (v_1^\top, \ldots, v_n^\top)^\top, \lambda_1 = (\lambda_{1,1}^\top, \ldots, \lambda_{1,n}^\top)^\top, \lambda_2 = (\lambda_{2,1}^\top, \ldots, \lambda_{2,n}^\top)^\top
$$

with $u_i, v_i, \lambda_{1,i}, \lambda_{2,i} \in \mathbb{R}^d$ stored locally at agent $i$, and communication is restricted to exchanging weighted neighbor-averaged values determined by the mixing matrix $W$. For convenience, for any stacked variable $z = (z_1^\top, \ldots, z_n^\top)^\top$ we use the notation $\tilde{z}_i^{(t)} = \sum_{j \in \mathcal{N}_i} W_{ij} z_j^{(t)}$.

A key structural feature of the symmetric ADMM updates is that the quadratic term $\widetilde{W}^\top \widetilde{W}$ in equation 5 requires 2-hop neighbor information. Consequently, one full iteration necessarily involves *two distinct rounds of communication*, and the communication pattern must therefore be scheduled with care. Our design follows two ordered principles: (1) achieve the minimal number of rounds by **restricting communication to exactly two rounds per iteration**; (2) within each round, **minimize the amount of transmitted data**.

The first principle is enforced because $\lambda_1^{(t+\frac{1}{2})}$ requires aggregated $u^{(t+1)}$, and $\lambda_2^{(t+1)}$ requires aggregated $v^{(t+1)}$, fixing the two communication rounds between these updates. Following this structure,

we organize each iteration into two update groups—Group 1 and Group 2—separated by two communication rounds. To satisfy the second principle, we transmit carefully chosen dual-variable combinations rather than raw primal variables, reducing the communication in each round to only two $d$-dimensional vectors and achieving the minimal cost compatible with the symmetric ADMM structure. The overall communication–computation workflow is described as follows:

**Group 1 update and Communication 1.** At iteration $t$, agent $i$ begins with its current local variables $(u_i^{(t)}, v_i^{(t)}, \lambda_{1,i}^{(t)}, \lambda_{2,i}^{(t-\frac{1}{2})})$ and the cached mixed values $\tilde{v}_i^{(t)}$ and $\tilde{b}_i^{(t)}$ from the previous iteration. In Group 1, agent $i$ first updates $\lambda_{2,i}^{(t)}$, then computes $u_i^{(t+1)}$ via the proximal operator of $f_i$, and finally forms the intermediate dual iterate $\lambda_{2,i}^{(t+\frac{1}{2})}$.

At this point, the first communication round occurs. Instead of transmitting all primal variables, each agent constructs the auxiliary message $a_i^{(t+1)} = \lambda_{2,i}^{(t+\frac{1}{2})} + \frac{1}{r}\left(\lambda_{2,i}^{(t+\frac{1}{2})} - \lambda_{2,i}^{(t)}\right)$, and broadcasts the pair $(u_i^{(t+1)}, a_i^{(t+1)})$ to its neighbors. The received messages are aggregated to provide all mixed information needed for the Group 2 updates.

**Group 2 update and Communication 2.** Using $\tilde{u}_i^{(t+1)}$ and $\tilde{a}_i^{(t+1)}$, agent $i$ performs the Group 2 update: it updates $\lambda_{1,i}^{(t+\frac{1}{2})}$, computes $v_i^{(t+1)}$ via the proximal operator of $g_i$, and then forms $\lambda_{1,i}^{(t+1)}$.

In the second communication round, each agent constructs $b_i^{(t+1)} = 2\lambda_{1,i}^{(t+1)} - \lambda_{1,i}^{(t+\frac{1}{2})}$, and broadcasts $(v_i^{(t+1)}, b_i^{(t+1)})$ to its neighbors, and received messages become inputs for the next Group 1 update.

Each iteration of DS-ADMM therefore consists of two local update groups and two neighbor-communication rounds. Information from one group is not used directly in its own update but enables the update of the other group, creating a feedback structure. This interleaving induces a coupled communication-update mechanism where each block drives progress in the other.

Furthermore, this structure makes symmetric ADMM not just suitable but essential: it accelerates convergence without increasing communication, and leads to a clean, symmetric algorithm. To our knowledge, this tightly coupled update-communication framework is novel in the decentralized optimization literature.

## 3.4 ALGORITHM

The resulting procedure is summarized in Algorithm 1 and illustrated in Figure 1, which we refer to as *Double-Communication Symmetric ADMM (DS-ADMM)*. The step-size parameters are set as $0 < r \le 1$ and $s = 1$.

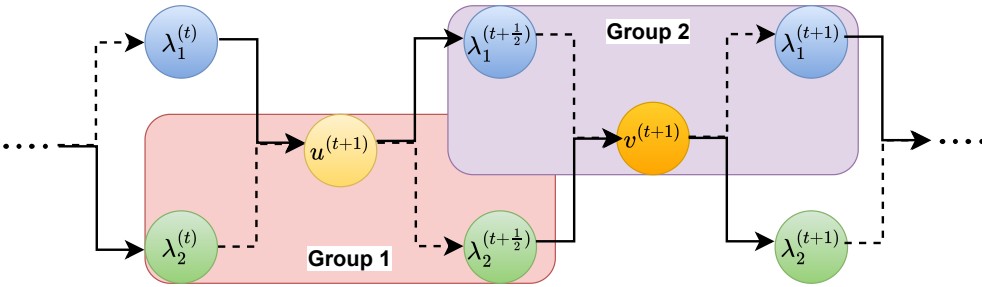

Figure 1: Illustration of one iteration of the proposed two-group symmetric decentralized ADMM scheme. Solid arrows denote *inter-group communication*, i.e., information exchanges that require communication across groups, while dashed arrows indicate *intra-group communication* performed locally within each group.

---

**Algorithm 1** Double-Communication Symmetric ADMM (DS-ADMM)

---

1: **Initialize:** $u_i^{(0)} = v_i^{(0)} = \lambda_{1,i}^{(0)} = \lambda_{2,i}^{(-\frac{1}{2})} = 0$ for all $i \in \{1, \dots, n\}$, mixing matrix $W \in \mathbb{R}^{n \times n}$, step sizes $r > 0$, augmented Lagrangian parameter $\beta > 0$ and proximal term parameter $\tau > 0$.
2: **repeat**
3:     **[Group 1 update]**
$$\lambda_{2,i}^{(t)} = \lambda_{2,i}^{(t-\frac{1}{2})} - s\beta(u_i^{(t)} - \tilde{v}_i^{(t)})$$
$$u_i^{(t+1)} = \text{Prox}_{\frac{1}{\beta(2+\tau)} f_i} \left( \frac{1}{2+\tau}(\tilde{v}_i^{(t)} + (1+\tau)u_i^{(t)}) + \frac{1}{(2+\tau)\beta}(\tilde{b}_i^{(t)} + \lambda_{2,i}^{(t)}) \right)$$
$$\lambda_{2,i}^{(t+\frac{1}{2})} = \lambda_{2,i}^{(t)} - r\beta(u_i^{(t+1)} - \tilde{v}_i^{(t)})$$
4:     **[Communication 1 ]** Transmit $a_i^{(t+1)} = \lambda_{2,i}^{(t+\frac{1}{2})} + \frac{1}{r}(\lambda_{2,i}^{(t+\frac{1}{2})} - \lambda_{2,i}^{(t)})$ and $u_i^{(t+1)}$.
5:     **[Group 2 update]**
$$\lambda_{1,i}^{(t+\frac{1}{2})} = \lambda_{1,i}^{(t)} - r\beta(\tilde{u}_i^{(t+1)} - v_i^{(t)})$$
$$v_i^{(t+1)} = \text{Prox}_{\frac{1}{\beta(2+\tau)} g_i} \left( \frac{1}{2+\tau}(\tilde{u}_i^{(t+1)} + (1+\tau)v_i^{(t)}) - \frac{1}{(2+\tau)\beta}(\lambda_{1,i}^{(t+\frac{1}{2})} + \tilde{a}_i^{(t+1)}) \right)$$
$$\lambda_{1,i}^{(t+1)} = \lambda_{1,i}^{(t)} - \beta(\tilde{u}_i^{(t+1)} - v_i^{(t+1)})$$
6:     **[Communication 2]** Transmit $v_i^{(t+1)}$ and $b_i^{(t+1)} = 2\lambda_{1,i}^{(t+1)} - \lambda_{1,i}^{(t+\frac{1}{2})}$.
7: **until** convergence criterion is satisfied

---

# 4 CONVERGENCE ANALYSIS

In this section, we analyze the convergence properties of the proposed decentralized algorithm. As it is a direct application of symmetric ADMM with proximal terms, several convergence results follow from existing literature. We further establish linear convergence under specific conditions that are mild yet broadly applicable to machine learning problems.

To facilitate the analysis, we define the block matrix:

$$H = \begin{pmatrix} Q & & \\ & Q + \frac{1}{r+1}\beta B^\top B & -\frac{r}{r+1}B^\top \\ & -\frac{r}{r+1}B & \frac{1}{\beta(r+1)}I \end{pmatrix}, \tag{6}$$

which is positive definite. We also define the concatenated variable $\theta = (u^\top, v^\top, \lambda^\top)^\top \in \mathbb{R}^{4nd}$.

## 4.1 GENERAL SUBLINEAR CONVERGENCE

Theorems 3.3 and 4.2 of (Gu et al., 2015) imply that the proposed algorithm enjoys a general sublinear convergence rate $\mathcal{O}(1/t)$ without requiring strong assumptions on the objective functions.

**Theorem 1.** *Let $\{\theta^{(t)}\}$ be the sequence generated by DS-ADMM. Then $\{\theta^{(t)}\}$ converges to a solution point $\theta^\infty$, and the following non-ergodic sublinear rate holds:*

$$\|\theta^{(t)} - \theta^{(t+1)}\|^2 \leq \frac{1}{\beta\tau(t+1)} \cdot \frac{1+r}{1-r} \left( \|\theta^{(1)} - \theta^{(0)}\|_H^2 + \|v^{(1)} - v^{(0)}\|_Q^2 \right). \tag{7}$$

**Remark 2.** *This sublinear convergence result is independent of the underlying communication graph and mixing matrix. Thus, the algorithm is inherently robust to different network topologies in decentralized environments.*

## 4.2 LINEAR CONVERGENCE UNDER METRIC SUBREGULARITY

Although various results on linear convergence of ADMM and its variants exist (e.g., (Deng & Yin, 2016; Han et al., 2018; Yuan et al., 2020; Bai et al., 2019; Gu et al., 2015)), none directly apply to our algorithm. Nevertheless, we adapt ideas from these works to establish linear convergence under a standard regularity condition known as metric subregularity.

**Definition 1** (Metric Subregularity). *A set-valued map $\Psi : \mathbb{R}^n \rightrightarrows \mathbb{R}^q$ is said to be* metrically subregular *at $(\bar{x}, \bar{y}) \in \text{gph}(\Psi)$ with modulus $\kappa > 0$ if there exists $\epsilon > 0$ such that:*

$$\text{dist}(x, \Psi^{-1}(\bar{y})) \leq \kappa \cdot \text{dist}(\bar{y}, \Psi(x)), \quad \forall x \in \mathbb{B}_\epsilon(\bar{x}). \tag{8}$$

We consider the KKT mapping:

$$T_{\text{KKT}}(\theta) := \begin{pmatrix} \partial f(u) - A^\top \lambda \\ \partial g(v) + B^\top \lambda \\ Au - Bv \end{pmatrix}, \tag{9}$$

and solution set $\Omega^* := \{\theta \mid 0 \in T_{\text{KKT}}(\theta)\}$.

Under the above framework, we are now in position to state the following linear convergence theorem which established a Q-linear rate of distance to the solution set, and a R-linear rate of suboptimality. The proof is deferred to Appendix C.

**Theorem 2.** *Suppose $T_{\text{KKT}}$ is metrically subregular at $(\bar{\theta}, 0)$ with modulus $c$ for any $\bar{\theta} \in \Omega^*$. Then the sequence $\{\theta^{(t)}\}$ generated by DS-ADMM converges Q-linearly to $\Omega^*$, i.e., there exist integer $T > 0$ and constant $\epsilon > 0$ such that for all $t > T$:*

$$\text{dist}_H^2(\theta^{(t+1)}, \Omega^*) \leq \frac{1}{1 + \epsilon} \cdot \text{dist}_H^2(\theta^{(t)}, \Omega^*), \tag{10}$$

*where*

$$\epsilon = \frac{\phi}{c^2 \delta \zeta} > 0, \quad \phi = \min\left\{2\beta\rho, \frac{1 - r}{\beta}\right\}, \tag{11}$$

*and*

$$\delta = \max\left\{6r^2 + \frac{2}{\beta^2}, 12\beta^2 + 4 + (\tau\beta)^2, 3(\tau\beta)^2\right\} \quad \zeta = \frac{2r^2\beta^2 + 1}{\beta(r + 1)} + (2 + \tau - r)\beta. \tag{12}$$

*Also the suboptimality converges R-linearly, which means there exists $l > 0$:*

$$|f(u^{(t)}) + g(v^{(t)}) - f(u^\infty) + g(v^\infty)| \leq lq^t, \quad q = \sqrt{\frac{1}{1 + \epsilon}}. \tag{13}$$

The linear convergence rate clearly depends on the algorithmic parameters $\tau$, $r$, $\beta$, and the structure of the mixing matrix $W$. In particular, a larger value of $\rho$, which reflects better network connectivity, leads to a faster convergence rate.

### 4.3 SUFFICIENT CONDITIONS FOR METRIC SUBREGULARITY

First we state the important definition of PLQ functions:

**Definition 2.** *A function $f : \mathbb{R}^n \to \mathbb{R} \cup \{+\infty\}$ is piecewise linear-quadratic (PLQ) if it is quadratic on a finite union of polyhedral regions:*

$$f(x) = \frac{1}{2}x^\top Q x + c^\top x + r. \tag{14}$$

Many loss and regularization terms used in machine learning are PLQ, including the $\ell_1$ and $\ell_2$ norm, hinge loss, squared loss, and elastic net.

The following proposition gives a characterization of the metric subregularity of $T_{\text{KKT}}$. Each case is justified by different results from the literature: Theorem 46 and Theorem 60 of (Yuan et al., 2020) support the first condition, Robinson's continuity property (Robinson, 1981) establishes the second, and Lemma 4 of (Latafat et al., 2019) together with Theorem 60 of (Yuan et al., 2020) imply the third.

**Proposition 3.** *The KKT mapping $T_{\text{KKT}}$ is metrically subregular at $(\bar{\theta}, 0)$ for any $\bar{\theta} \in \Theta^*$ under any of the following conditions: (i) each $f_i$ is smooth and strongly convex, and each $g_i$ is PLQ; (ii) all $f_i$ and $g_i$ are PLQ; (iii) all $f_i$ and $g_i$ are smooth and strongly convex.*

Therefore, DS-ADMM achieves linear convergence across a wide range of practical decentralized optimization problems, including Lasso, logistic regression, SVM classification and other models frequently encountered in machine learning.

## 5 NUMERICAL EXPERIMENTS

We evaluate the proposed DS-ADMM algorithm on two representative decentralized composite optimization tasks: Lasso regression and $\ell_2$-regularized SVM classification. All experiments use $n = 30$ agents connected via a random graph with edge probability $p = 0.5$. The Metropolis–Hastings mixing matrix is constructed following Appendix A, and data are evenly partitioned among agents. The global regularizer is split uniformly as $g_i(x) = \frac{1}{n}g(x)$. Comparisons are made against four representative methods: Decentralized Proximal ADMM (Wang et al., 2018), PG-EXTRA (Shi et al., 2015b), NIDS (Li et al., 2019) and ProxMudag (Ye et al., 2023). All experiments are conducted on a machine equipped with an Intel Core i7-1260P CPU and 16GB RAM.

All adaptive parameters of baseline methods are tuned for best performance. For DS-ADMM, we use fixed step sizes $(r, s) = (0.99, 1)$ and proximal coefficient $\tau = 0.01$. Suboptimality is measured as $F(\bar{u}^{(t)}) - F(u^\star)$, where $u^\star$ is a high-accuracy centralized solution.

**Lasso Regression.** Using the a9a dataset from the LIBSVM repository (Chang & Lin, 2011) with

$$f_i(x) = \frac{1}{2m}\|A_i x - b_i\|^2, \qquad g_i(x) = \frac{\lambda}{n}\|x\|_1, \quad \lambda = \frac{1}{m}.$$

**SVM Classification.** Using a1a, each agent solves

$$f_i(x) = \frac{1}{m}\sum_{j \in S_i}\max(0, 1 - b_j a_j^\top x), \qquad g_i(x) = \frac{\lambda}{2n}\|x\|_2^2.$$

Note that ProxMudag is excluded in the SVM classification task as its formulation requires globally coupled nonsmooth terms. Across both tasks, DS-ADMM consistently converges faster and requires significantly fewer communication rounds. Additional results and parameter sensitivity analysis are reported in Appendix D.

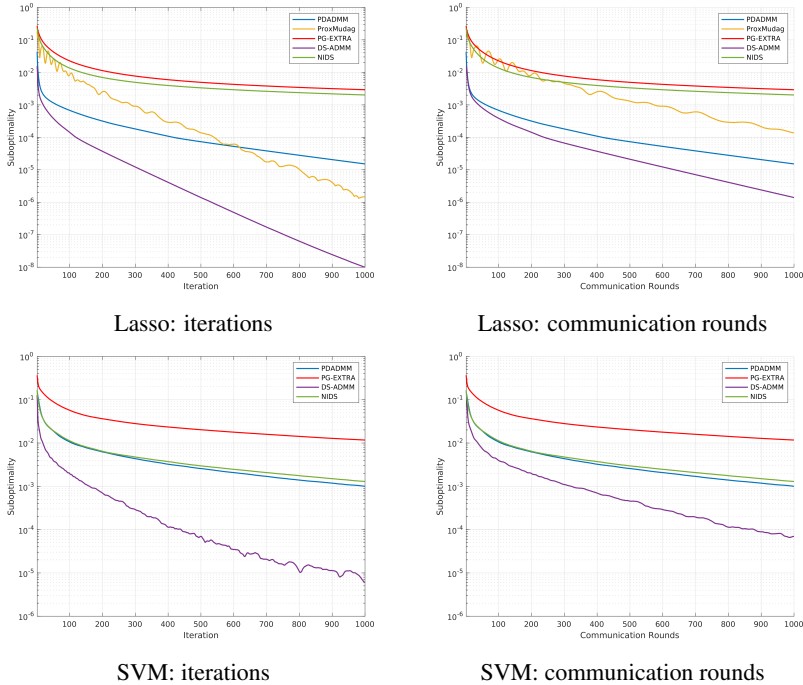

Figure 2: Comparison of DS-ADMM with baseline methods on Lasso and SVM.

## 6 CONCLUSION

We proposed DS-ADMM, a fully decentralized algorithm for composite optimization based on the symmetric ADMM framework. The method integrates a novel communication structure that structures

double communication per iteration without increasing overhead, while maintaining a symmetric update pattern across variable blocks. Theoretical analysis established both sublinear convergence under standard conditions, with linear rates guaranteed by metric subregularity of the KKT mapping. Our algorithm accommodates general nonsmooth regularization and is broadly applicable to various machine learning tasks. Extensive numerical experiments demonstrate that DS-ADMM outperforms existing decentralized methods in both iteration and communication efficiency.

## ACKNOWLEDGMENTS

This research is partially supported by the National Natural Science Foundation of China (12231017, 72171216) and the National Key R&D Program of China (2022YFA1003803). We gratefully acknowledge the insightful comments and suggestions provided by the anonymous reviewers.

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

# Appendix

## A MIXING MATRIX BASED ON METROPOLIS-HASTINGS WEIGHT

A desired mixing matrix can be constructed using Metropolis-Hastings weights, where $d_i = |\mathcal{N}_i|$ is the degree of node $i$:

$$
\mathcal{W}_{ij} = \begin{cases} \frac{1}{1+\max\{d_i, d_j\}}, & \text{if } (i,j) \in E, \\ 0, & \text{if } (i,j) \notin E \text{ and } j \neq i, \\ 1 - \sum_{l \in \mathcal{N}_i} \mathcal{W}_{il}, & \text{if } j = i, \end{cases}
$$

## B GLOBAL FORM OF UPDATE

According to the update rule of Symmetric ADMM and our linear constraint formulation, a global $t$-th iteration can be written in the below equation:

$$
\begin{cases} u^{(t+1)} = \arg\min_{u \in \mathbb{R}^{nd}} f(u) - \langle \lambda^{(t)}, Au \rangle + \frac{\beta}{2} \|Au - Bv^{(t)}\|^2 + \frac{1}{2} \|u - u^{(t)}\|_Q^2, \\ \lambda^{(t+\frac{1}{2})} = \lambda^{(t)} - r\beta(Au^{(t+1)} - Bv^{(t)}) \\ v^{(t+1)} = \arg\min_{v \in \mathbb{R}^{nd}} g(v) + \langle \lambda^{(t+\frac{1}{2})}, Bv \rangle + \frac{\beta}{2} \|Au^{(t+1)} - Bv\|^2 + \frac{1}{2} \|v - v^{(t)}\|_Q^2, \\ \lambda^{(t+1)} = \lambda^{(t+\frac{1}{2})} - s\beta(Au^{(t+1)} - Bv^{(t+1)}). \end{cases} \tag{15}
$$

Then we decompose the Lagrange multiplier $\lambda$, which yields the below equation:

$$
\begin{cases} u^{(t+1)} = \arg\min_{u \in \mathbb{R}^{nd}} f(u) - \langle \lambda_1^{(t)}, \widetilde{W}u \rangle - \langle \lambda_2^{(t)}, u \rangle + \frac{\beta}{2} \|\widetilde{W}u - v^{(t)}\|^2 + \frac{\beta}{2} \|u - \widetilde{W}v^{(t)}\|^2 \\ \qquad + \frac{1}{2} \|u - u^{(t)}\|_Q^2, \\ \lambda_1^{(t+\frac{1}{2})} = \lambda_1^{(t)} - r\beta(\widetilde{W}u^{(t+1)} - v^{(t)}), \quad \lambda_2^{(t+\frac{1}{2})} = \lambda_2^{(t)} - r\beta(u^{(t+1)} - \widetilde{W}v^{(t)}), \\ v^{(t+1)} = \arg\min_{v \in \mathbb{R}^{nd}} g(v) + \langle \lambda_1^{(t+\frac{1}{2})}, v \rangle + \langle \lambda_2^{(t+\frac{1}{2})}, \widetilde{W}v \rangle + \frac{\beta}{2} \|\widetilde{W}u^{(t+1)} - v\|^2 \\ \qquad + \frac{\beta}{2} \|u^{(t+1)} - \widetilde{W}v\|^2 + \frac{1}{2} \|v - v^{(t)}\|_Q^2, \\ \lambda_1^{(t+1)} = \lambda_1^{(t+\frac{1}{2})} - s\beta(\widetilde{W}u^{(t+1)} - v^{(t+1)}), \quad \lambda_2^{(t+1)} = \lambda_2^{(t+\frac{1}{2})} - s\beta(u^{(t+1)} - \widetilde{W}v^{(t+1)}). \end{cases} \tag{16}
$$

And it can be transformed into the following version ready for decentralized update:

$$
\begin{cases} u^{(t+1)} = \arg\min_{u \in \mathbb{R}^{nd}} f(u) - \langle \widetilde{W}\lambda_1^{(t)} + \lambda_2^{(t)} + \beta(2\widetilde{W}v^{(t)} + (1+\tau)u^{(t)} - \widetilde{W}^\top\widetilde{W}u^{(t)}), u \rangle \\ \qquad + \beta(1 + \frac{\tau}{2}) \|u\|^2 \\ \lambda_1^{(t+\frac{1}{2})} = \lambda_1^{(t)} - r\beta(\widetilde{W}u^{(t+1)} - v^{(t)}), \quad \lambda_2^{(t+\frac{1}{2})} = \lambda_2^{(t)} - r\beta(u^{(t+1)} - \widetilde{W}v^{(t)}), \\ v^{(t+1)} = \arg\min_{v \in \mathbb{R}^{nd}} g(v) - \langle \beta(2\widetilde{W}u^{(t+1)} + (1+\tau)v^{(t)} - \widetilde{W}^\top\widetilde{W}v^{(t)}) - (\lambda_1^{(t+\frac{1}{2})} + \\ \qquad \widetilde{W}\lambda_2^{(t+\frac{1}{2})}), v \rangle + \beta(1 + \frac{\tau}{2}) \|v\|^2 \\ \lambda_1^{(t+1)} = \lambda_1^{(t+\frac{1}{2})} - s\beta(\widetilde{W}u^{(t+1)} - v^{(t+1)}), \quad \lambda_2^{(t+1)} = \lambda_2^{(t+\frac{1}{2})} - s\beta(u^{(t+1)} - \widetilde{W}v^{(t+1)}). \end{cases} \tag{17}
$$

Rearranging the terms and using the facts that $\widetilde{W}v^{(t)} - \widetilde{W}^\top\widetilde{W}u^{(t)} = \frac{1}{s\beta}\widetilde{W}(\lambda_1^{(t)} - \lambda_1^{(t-\frac{1}{2})})$ and $\widetilde{W}u^{(t+1)} - \widetilde{W}^\top\widetilde{W}v^{(t)} = -\frac{1}{r\beta}\widetilde{W}(\lambda_2^{(t+\frac{1}{2})} - \lambda_1^{(t)})$ in the above equation, we get:

$$\begin{cases} u^{(t+1)} = \arg\min_{u \in \mathbb{R}^{nd}} f(u) - \langle \widetilde{W}\lambda_1^{(t)} + \lambda_2^{(t)} + \beta(\widetilde{W}v^{(t)} + (1+\tau)u^{(t)}) + \frac{1}{s}\widetilde{W}(\lambda_1^{(t)} - \lambda_1^{(t-\frac{1}{2})}), u\rangle \\ \qquad + \beta(1 + \frac{\tau}{2})\|u\|^2 \\ \lambda_1^{(t+\frac{1}{2})} = \lambda_1^{(t)} - r\beta(\widetilde{W}u^{(t+1)} - v^{(t)}), \quad \lambda_2^{(t+\frac{1}{2})} = \lambda_2^{(t)} - r\beta(u^{(t+1)} - \widetilde{W}v^{(t)}), \\ v^{(t+1)} = \arg\min_{v \in \mathbb{R}^{nd}} g(v) - \langle \beta(\widetilde{W}u^{(t+1)} + (1+\tau)v^{(t)}) - (\lambda_1^{(t+\frac{1}{2})} + \widetilde{W}\lambda_2^{(t+\frac{1}{2})}) \\ \qquad + \frac{1}{r}\widetilde{W}(\lambda_2^{(t+\frac{1}{2})} - \lambda_1^{(t)})), v\rangle + \beta(1 + \frac{\tau}{2})\|v\|^2 \\ \lambda_1^{(t+1)} = \lambda_1^{(t+\frac{1}{2})} - s\beta(\widetilde{W}u^{(t+1)} - v^{(t+1)}), \quad \lambda_2^{(t+1)} = \lambda_2^{(t+\frac{1}{2})} - s\beta(u^{(t+1)} - \widetilde{W}v^{(t+1)}). \end{cases} \tag{18}$$

By our previous definition of $a^{(t)} = \lambda_2^{(t+\frac{1}{2})} + \frac{1}{r}(\lambda_2^{(t+\frac{1}{2})} - \lambda_2^{(t)})$ and $b^{(t)} = \lambda_1^{(t)} + \frac{1}{s}(\lambda_1^{(t)} - \lambda_1^{(t-\frac{1}{2})})$, we get

$$\begin{cases} u^{(t+1)} = \arg\min_{u \in \mathbb{R}^{nd}} f(u) - \langle \widetilde{W}a^{(t)} + \lambda_2^{(t)} + \beta(\widetilde{W}v^{(t)} + (1+\tau)u^{(t)}), u\rangle + \beta(1 + \frac{\tau}{2})\|u\|^2 \\ \lambda_1^{(t+\frac{1}{2})} = \lambda_1^{(t)} - r\beta(\widetilde{W}u^{(t+1)} - v^{(t)}), \quad \lambda_2^{(t+\frac{1}{2})} = \lambda_2^{(t)} - r\beta(u^{(t+1)} - \widetilde{W}v^{(t)}), \\ v^{(t+1)} = \arg\min_{v \in \mathbb{R}^{nd}} g(v) - \langle \beta(\widetilde{W}u^{(t+1)} + (1+\tau)v^{(t)}) - (\lambda_1^{(t+\frac{1}{2})} + \widetilde{W}b^{(t)}), v\rangle + \beta(1 + \frac{\tau}{2})\|v\|^2 \\ \lambda_1^{(t+1)} = \lambda_1^{(t+\frac{1}{2})} - s\beta(\widetilde{W}u^{(t+1)} - v^{(t+1)}), \quad \lambda_2^{(t+1)} = \lambda_2^{(t+\frac{1}{2})} - s\beta(u^{(t+1)} - \widetilde{W}v^{(t+1)}). \end{cases} \tag{19}$$

## C    PROOF OF THEOREM 2

The proof is based on the notations of global form equation 7. First, we denote vectors

$$z = \begin{pmatrix} u \\ v \end{pmatrix}, \quad \theta = \begin{pmatrix} u \\ v \\ \lambda \end{pmatrix}, \quad F(\theta) = \begin{pmatrix} -A^\top\lambda \\ B^\top\lambda \\ Au - Bv \end{pmatrix} \tag{20}$$

$$\widetilde{u}^{(t)} = u^{(t+1)}, \quad \widetilde{v}^{(t)} = v^{(t+1)}, \quad \widetilde{\lambda}^{(t)} = \lambda^{(t)} - \beta\left(Au^{(t+1)} - Bv^{(t+1)}\right), \tag{21}$$

$$\widetilde{z}^{(t)} = \begin{pmatrix} \widetilde{u}^{(t)} \\ \widetilde{v}^{(t)} \end{pmatrix}, \quad \widetilde{\theta}^{(t)} = \begin{pmatrix} \widetilde{u}^{(t)} \\ \widetilde{v}^{(t)} \\ \widetilde{\lambda}^{(t)}, \end{pmatrix}, \quad h(z) = f(x) + g(y) \tag{22}$$

and matrices

$$S = \begin{pmatrix} Q & & \\ & Q + \beta B^\top B & -rB^\top \\ & B & \frac{1}{\beta}I \end{pmatrix}, M = \begin{pmatrix} I & & \\ & I & \\ & -\beta B & (1+r)I \end{pmatrix}, \tag{23}$$

$$H = \begin{pmatrix} Q & & \\ & Q + \frac{1}{r+1}\beta B^\top B & -\frac{r}{r+1}B^\top \\ & -\frac{r}{r+1}B & \frac{1}{\beta(r+1)}I \end{pmatrix}, G = \begin{pmatrix} Q & & \\ & Q & \\ & & \frac{1-r}{\beta}I \end{pmatrix}, \tag{24}$$

It is easy to verify the following properties:

$$\theta^{(t+1)} = \theta^{(t)} - M(\theta^{(t)} - \widetilde{\theta}^{(t)}). \tag{25}$$

$$G = S + S^\top - M^\top S, \quad H = SM^{-1}. \tag{26}$$

The following lemma characterizes the eigenvalue property of $H$ and $G$:

**Lemma 1.**
$$\lambda_{\max}(H) \le \zeta, \quad \lambda_{\min}(G) \ge 2(1-r)\rho \tag{27}$$

*Proof.*
$$H = \begin{pmatrix} Q & & \\ & Q + (1-r)\beta B^\top B + \frac{r^2}{r+1}\beta B^\top B & -\frac{r}{r+1}B^\top \\ & -\frac{r}{r+1}B & \frac{1}{\beta(r+1)}I \end{pmatrix} \tag{28}$$

so it is easy to verify that

$$\lambda_{\max}(H) \le \lambda_{\max}(Q + (1-r)\beta B^\top B) + \lambda_{\max}\Big(\frac{1}{\beta(r+1)}\begin{pmatrix} r\beta B^\top \\ -I \end{pmatrix}\begin{pmatrix} r\beta B^\top \\ -I \end{pmatrix}^\top\Big) \tag{29}$$

while

$$\lambda_{\max}(Q + (1-r)\beta B^\top B) = \lambda_{\max}(\beta(1+\tau)I - \beta\widetilde{W}^\top\widetilde{W} + (1-r)\beta(I + \widetilde{W}^\top\widetilde{W})) \le (2+\tau-r)\beta, \tag{30}$$

$$\lambda_{\max}\Big(\frac{1}{\beta(r+1)}\begin{pmatrix} r\beta B^\top \\ -I \end{pmatrix}\begin{pmatrix} r\beta B^\top \\ -I \end{pmatrix}^\top\Big) = \frac{1}{\beta(r+1)}\lambda_{\max}((r\beta)^2(I + \widetilde{W}^\top\widetilde{W}) + I) = \frac{2r^2\beta^2 + 1}{\beta(r+1)}, \tag{31}$$

combining above, we get $\lambda_{\max}(H) \le \zeta$. Also, we have

$$\lambda_{\min}(G) = \min\{\lambda_{\min}(Q), \frac{1-r}{\beta}\} \ge \min\{2\beta\rho, \frac{1-r}{\beta}\} = \phi. \tag{32}$$

$\square$

This equivalent form of solution set $\Omega^*$ follows from (He et al., 2016) by the definition of subdifferentials:

$$\Omega^* = \bigcap_{\theta \in \mathbb{R}^{4nd}} \left\{\widehat{\theta} \,\Big|\, h(z) - h(\widehat{z}) + \langle\theta - \widehat{\theta}, F(\theta)\rangle \ge 0\right\}. \tag{33}$$

Then we give the following important lemma, whose proof follows directly from the proof of Theorem 2 and Theorem 3 in (Bai et al., 2018) and the above form of solution set:

**Lemma 2.**
$$\left\|\theta^{(t+1)} - \theta^*\right\|_H^2 \le \left\|\theta^{(t)} - \theta^*\right\|_H^2 - \left\|\theta^{(t)} - \widetilde{\theta}^{(t)}\right\|_G^2, \quad \forall\theta^* \in \Omega^*, \tag{34}$$

Let us revise the KKT mapping

$$T_{\text{KKT}}(\theta) := \begin{pmatrix} \partial f(u) - A^\top\lambda \\ \partial g(v) + B^\top\lambda \\ Au - Bv \end{pmatrix}. \tag{35}$$

The following lemma bounds the distance through $G$-norm.

**Lemma 3.** *The sequences $\{\theta^{(t)}\}$ and $\{\widetilde{\theta}^{(t)}\}$ satisfy*

$$\text{dist}^2(0, T_{KKT}(\theta^{(t+1)})) \le \frac{\delta}{\phi}\left\|\theta^{(t)} - \widetilde{\theta}^{(t)}\right\|_G^2,$$

*Proof.* By Proposition 2.1 in (He et al., 2016) and our notations, we derive the following inequality characterizing the subproblems:

$$f(u) - f(\widetilde{u}^{(t)}) + \left\langle u - \widetilde{u}^{(t)}, -A^\top\widetilde{\lambda}^{(t)} + Q(\widetilde{u}^{(t)} - u^{(t)})\right\rangle \ge 0 \tag{36}$$

$$g(v) - g(\widetilde{v}^{(t)}) + \left\langle v - \widetilde{v}^{(t)}, -B^\top\widetilde{\lambda}^{(t)} - rB^\top(\widetilde{\lambda}^{(t)} - \lambda^{(t)}) + (Q + \beta B^\top B)(\widetilde{v}^{(t)} - v^{(t)})\right\rangle \ge 0 \tag{37}$$

It is obvious that

$$\text{dist}^2(0, T_{\text{KKT}}(\theta)) = \text{dist}^2(0, T_1(\theta)) + \text{dist}^2(0, T_2(\theta)) + \text{dist}^2(0, T_3(\theta)) \tag{38}$$

where $T_1(\theta) = \partial f(u) - A^\top \lambda$, $T_2(\theta) = \partial g(v) + B^\top \lambda$, $T_3(\theta) = Au - Bv$.
From the above two inequalities, we have the following:

$$
\begin{aligned}
\operatorname{dist}(0, T_1(\theta^{(t+1)})) &\leq \left\| A^\top(\widetilde{\lambda}^{(t)} - \lambda^{(t+1)}) - Q(\widetilde{u}^{(t)} - u^{(t)}) \right\| \\
&= \left\| A^\top \left[ r(\lambda^{(t)} - \widetilde{\lambda}^{(t)}) - \beta B(v^{(t)} - \widetilde{v}^{(t)}) \right] - Q(\widetilde{u}^{(t)} - u^{(t)}) \right\|,
\end{aligned}
\tag{39}
$$

where the second equality uses the update rule

$$
\begin{aligned}
\lambda^{(t+1)} &= \lambda^{(t+\frac{1}{2})} - \beta(Au^{(t+1)} - Bv^{(t+1)}) \\
&= \lambda^{(t+\frac{1}{2})} - \beta(Au^{(t+1)} - Bv^{(t+1)}) + \beta B(v^{(t)} - v^{(t+1)}) \\
&= \lambda^k - (r+1)(\lambda^k - \widetilde{\lambda}^k) + \beta B(v^{(t)} - v^{(t+1)})
\end{aligned}
\tag{40}
$$

similarly, we have:

$$
\begin{aligned}
\operatorname{dist}(0, T_2(\theta^{(t+1)})) &\leq \left\| B^\top(\widetilde{\lambda}^{(t)} - \lambda^{(t+1)}) - (Q + \beta B^\top B)(\widetilde{v}^{(t)} - v^{(t)}) + r B^\top(\widetilde{\lambda}^{(t)} - \lambda^{(t)}) \right\| \\
&= \left\| Q(\widetilde{v}^{(t)} - v^{(t)}) \right\|,
\end{aligned}
\tag{41}
$$

and finally

$$
\operatorname{dist}(0, T_3(\theta^{(t+1)})) = \left\| \frac{1}{\beta}(\lambda^{(t)} - \widetilde{\lambda}^{(t)}) - B(v^{(t)} - \widetilde{v}^{(t)}) \right\|.
\tag{42}
$$

Substituting equation 39 equation 41 equation 42 into equation 38 while using the definition of $A$, $B$, $Q$, we get

$$
\begin{aligned}
\operatorname{dist}^2(0, T_{\text{KKT}}(\theta^{(t+1)})) &\leq \left\| A^\top \left[ r(\lambda^{(t)} - \widetilde{\lambda}^{(t)}) - \beta B(v^{(t)} - \widetilde{v}^{(t)}) \right] - Q(\widetilde{u}^{(t)} - u^{(t)}) \right\|^2 \\
&\quad + \left\| Q(\widetilde{v}^{(t)} - v^{(t)}) \right\|^2 + \left\| \frac{1}{\beta}(\lambda^{(t)} - \widetilde{\lambda}^{(t)}) - B(v^{(t)} - \widetilde{v}^{(t)}) \right\|^2 \\
&\leq 3r^2 \left\| A^\top(\lambda^{(t)} - \widetilde{\lambda}^{(t)}) \right\|^2 + \frac{2}{\beta^2} \left\| \lambda^{(t)} - \widetilde{\lambda}^{(t)} \right\|^2 \\
&\quad + 3\beta^2 \left\| A^\top B(v^{(t)} - \widetilde{v}^{(t)}) \right\|^2 \\
&\quad + 2 \left\| B(v^{(t)} - \widetilde{v}^{(t)}) \right\|^2 + \left\| Q(v^{(t)} - \widetilde{v}^{(t)}) \right\|^2 + 3 \left\| Q(\widetilde{u}^{(t)} - u^{(t)}) \right\|^2 \\
&\leq (6r^2 + \frac{2}{\beta^2}) \left\| \lambda^{(t)} - \widetilde{\lambda}^{(t)} \right\|^2 + (12\beta^2 + 4 + (\tau\beta)^2) \left\| v^{(t)} - \widetilde{v}^{(t)} \right\|^2 \\
&\quad + 3(\tau\beta)^2 \left\| u^{(t)} - \widetilde{u}^{(t)} \right\|^2 \\
&\leq \delta \left\| \theta^{(t)} - \widetilde{\theta}^{(t)} \right\|^2 \leq \frac{\delta}{\phi} \left\| \theta^{(t)} - \widetilde{\theta}^{(t)} \right\|_G^2.
\end{aligned}
\tag{43}
$$

$\square$

Then we are able to prove the first part of Theorem 2:

*Proof.* Because $\Omega^*$ is a closed convex set, there exists a $\theta_t^* \in \Omega^*$ satisfying

$$
\operatorname{dist}_H(\theta^{(t)}, \Omega^*) = \left\| \theta^{(t)} - \theta_t^* \right\|_H.
\tag{44}
$$

Then, by the metric subregularity of the KKT mapping and the global convergence result, there exists $T > 0$, for any $t > T$ we have

$$\left\| \theta^{(t)} - \widetilde{\theta}^{(t)} \right\|_G \geq \sqrt{\frac{\phi}{\delta}} \mathrm{dist}(0, T_{\mathrm{KKT}}(\theta^{(t+1)}))$$

$$\geq \sqrt{\frac{\phi}{c^2\delta}} \, \mathrm{dist}(\theta^{(t+1)}, \Omega^*) \qquad (45)$$

$$\geq \sqrt{\frac{\phi}{c^2\delta\zeta}} \, \mathrm{dist}_H(\theta^{(t+1)}, \Omega^*).$$

So, we will have from the above inequality that

$$(1 + \epsilon) \, \mathrm{dist}_H^2(\theta^{(t+1)}, \Omega^*) \leq \left\| \theta^{(t+1)} - \theta_t^* \right\|_H^2 + \epsilon \, \mathrm{dist}_H^2(\theta^{(t+1)}, \Omega^*)$$

$$\leq \left\| \theta^{(t+1)} - \theta_t^* \right\|_H^2 + \left\| \theta^{(t)} - \widetilde{\theta}^{(t)} \right\|_G^2 \qquad (46)$$

$$\leq \left\| \theta^{(t)} - \theta_t^* \right\|_H^2 = \mathrm{dist}_H^2(\theta^{(t)}, \Omega^*)$$

and the main result is proved.

To prove the R-linear rate of suboptimality, first, from the same approach of proving Corollary 2.1 in (Bai et al., 2019) we are able to prove the R-linear rate of $\left\| \theta^{(t)} - \theta^\infty \right\|$. And a simple corollary from Lemma 2 is that $\left\| \theta^{(t)} - \tilde{\theta}^{(t)} \right\|$ and $\left\| Au^{(t+1)} - Bv^{(t+1)} \right\|$ also converges R-linearly.

Then we substitute $u^\infty$ and $v^\infty$ into equation 36 and equation 37:

$$f(u^\infty) + g(v^\infty) \geq f(\widetilde{u}^{(t)}) + g(\widetilde{v}^{(t)}) - \left\langle u^\infty - \widetilde{u}^{(t)}, -A^\top \widetilde{\lambda}^{(t)} + Q(\widetilde{u}^{(t)} - u^{(t)}) \right\rangle$$

$$- \left\langle v^\infty - \widetilde{v}^{(t)}, -B^\top \widetilde{\lambda}^{(t)} - rB^\top(\widetilde{\lambda}^{(t)} - \lambda^{(t)}) + (Q + \beta B^\top B)(\widetilde{v}^{(t)} - v^{(t)}) \right\rangle$$
$$(47)$$

and from the definition of solution set

$$f(u^{(t+1)}) + g(v^{(t+1)}) \geq f(u^\infty) + g(v^\infty) + \langle \lambda^\infty, Au^{(t+1)} - Bv^{(t+1)} \rangle. \qquad (48)$$

the above inequalities and the R-linear convergence of $\left\| \theta^{(t)} - \theta^\infty \right\|$, $\left\| \theta^{(t)} - \tilde{\theta}^{(t)} \right\|$, $\left\| Au^{(t+1)} - Bv^{(t+1)} \right\|$ lead to the R-linear convergence of suboptimality, which finishes the proof.

$\square$

## D ADDITIONAL EXPERIMENTAL DETAILS

### D.1 PARAMETER SENSITIVITY ANALYSIS

We summarize the hyperparameters used in the DS-ADMM algorithm and clarify their respective roles.

**Proximal parameter $\tau$.** The proximal term ensures that the matrix $Q$ is positive definite, which is required in order to establish linear convergence. Throughout our experiments, we set $\tau = 0.01$.

**Augmented Lagrangian parameter $\beta$.** The parameter $\beta$ has a major influence on the convergence speed. One may either adopt the adaptive tuning strategy proposed in (Boyd et al., 2011) or evaluate several candidate values and select the one that performs best empirically.

**Dual update step sizes $(r, s)$.** The convergence region and admissible ranges for the symmetric dual step sizes have been characterized in prior work (Bai et al., 2018). Following these theoretical guidelines, we fix $s = 1$ and $r = 0.99$, which consistently yield stable and efficient performance.

## D.2 ADDITIONAL EXPERIMENTAL RESULTS

We report experimental results on a 30-agent sparsely connected random graph with edge probability $p = 0.2$. As expected, all algorithms perform worse on this sparse topology, consistent with theoretical predictions. Nevertheless, our proposed DS-ADMM retains the best overall performance.

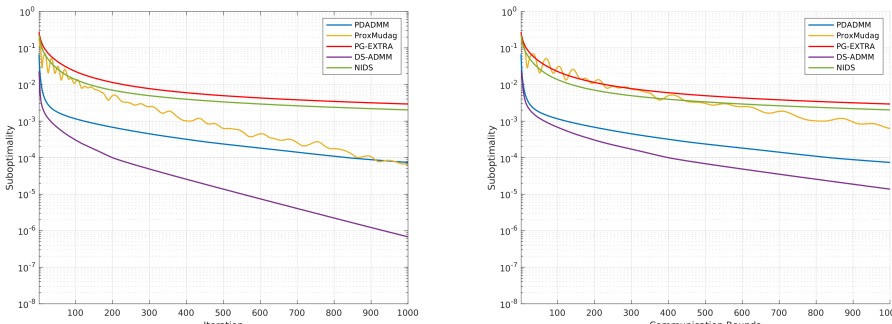

Figure 3: Lasso performance on the 30-agent sparse random graph ($p = 0.2$). Left: suboptimality vs. iterations. Right: suboptimality vs. communication rounds.

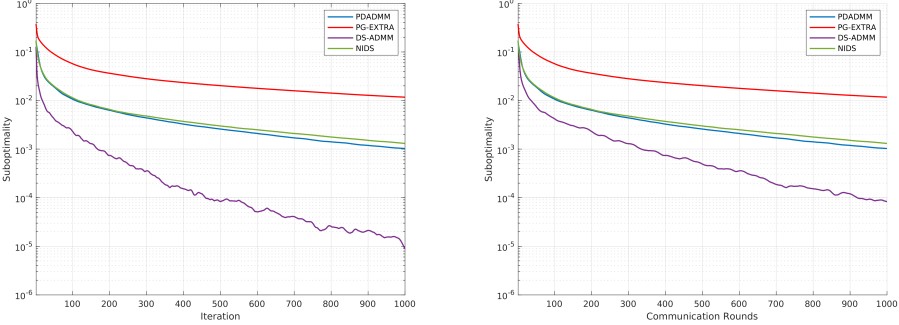

Figure 4: SVM performance on the 30-agent sparse random graph ($p = 0.2$). Left: suboptimality vs. iterations. Right: suboptimality vs. communication rounds.

We further evaluate the Lasso problem on a 100-agent network under two connectivity levels, with edge probabilities $p = 0.5$ and $p = 0.2$.

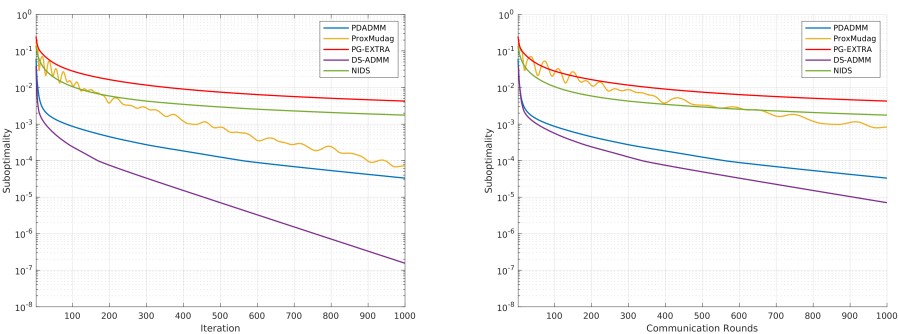

Figure 5: Lasso performance on the 100-agent random graph ($p = 0.5$). Left: suboptimality vs. iterations. Right: suboptimality vs. communication rounds.

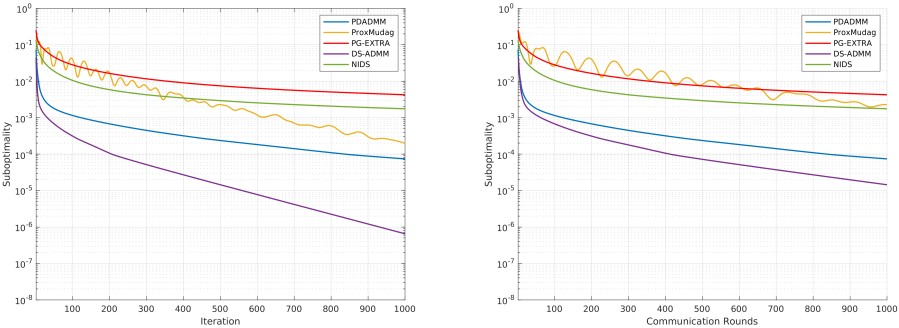

Figure 6: Lasso performance on the 100-agent random graph ($p = 0.2$). Left: suboptimality vs. iterations. Right: suboptimality vs. communication rounds.

Then we test performance when the dataset is randomly partitioned among agents, using $n = 30$ and $n = 100$ with graph edge probability $p = 0.5$.

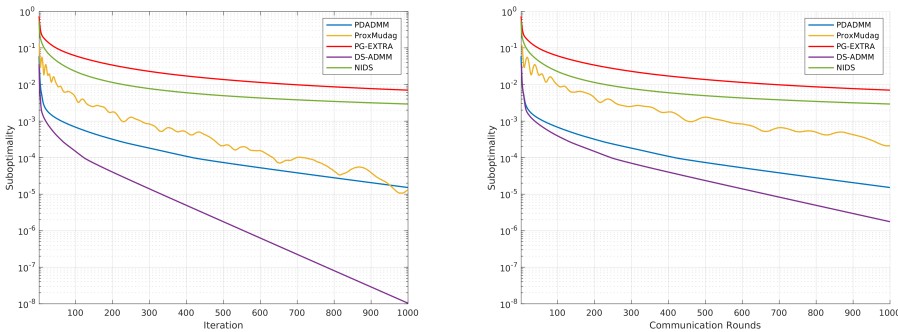

Figure 7: Lasso performance with random data partition on the 30-agent random graph ($p = 0.5$). Left: suboptimality vs. iterations. Right: suboptimality vs. communication rounds.

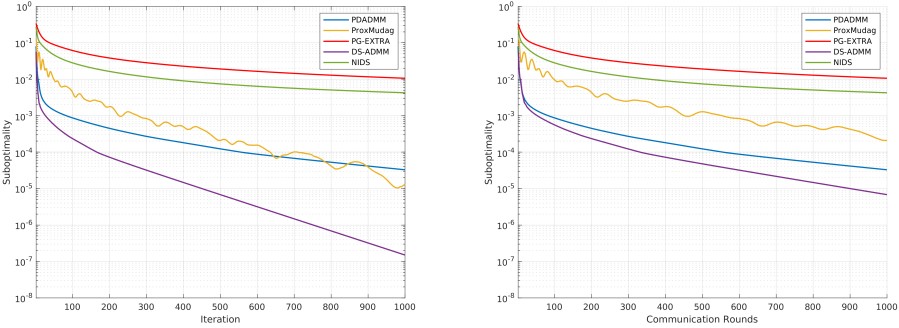

Figure 8: Lasso performance with random data partition on the 100-agent random graph ($p = 0.5$). Left: suboptimality vs. iterations. Right: suboptimality vs. communication rounds.

Finally, we test the performance on a ring network with 50 agents.

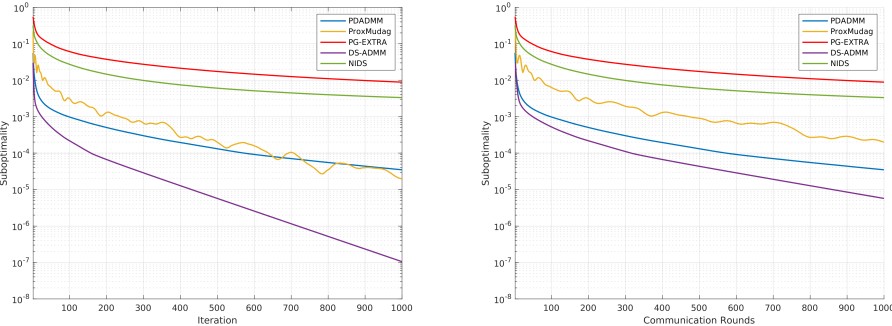

Figure 9: Lasso performance on the 50-agent ring topology. Left: suboptimality vs. iterations. Right: suboptimality vs. communication rounds.

## E    LLM USAGE

We used a large language model (LLM) to help polish the writing and enhance readability. The authors are solely responsible for the technical content, analysis, and conclusions.

