# OpenReview forum: "Communication-Efficient Decentralized Optimization via Double-Communication Symmetric ADMM"
_ICLR.cc/2026/Conference — ICLR 2026 Poster_

### Official Review · Reviewer_AAag · 2025-10-22

**Soundness:** 3
**Presentation:** 2
**Contribution:** 2
**Rating:** 4
**Confidence:** 3

**Summary:**

This paper proposes DS-ADMM, a decentralized optimization algorithm for composite problems that incorporates two communication rounds per iteration. The key innovation is a reformulation of consensus constraints that enables information exchange beyond immediate neighbors through a symmetric ADMM framework. The authors claim this approach reduces total communication cost despite increasing per-iteration communication. They provide convergence analysis showing sublinear rates generally and linear rates under metric subregularity conditions. Experiments on Lasso regression and SVM classification demonstrate superior performance compared to existing methods.

**Strengths:**

- The reformulation of consensus constraints using the symmetric form $u = \tilde{W} v, \tilde{W} u = v$ is creative and well-motivated. This differs meaningfully from prior work (Wang et al., 2018) and enables the application of Symmetric ADMM in a decentralized setting.
- The analysis in Section 3.3 carefully optimizes which variables to transmit and when. The identification that only two communication rounds are necessary and the strategic transmission of dual variables ($a_i^{(t)}$ and $b_i^{(t+1)}$) rather than primal variables shows careful algorithmic design.
- The paper provides both sublinear (Theorem 1) and linear convergence guarantees (Theorem 2). The characterization of sufficient conditions for metric subregularity (Proposition 4) covering PLQ functions and strongly convex cases is comprehensive and practically relevant.

**Weaknesses:**

- Limited Experimental Scope: Only two problem types (Lasso and SVM) and two relatively small datasets (a9a, a1a) are tested. Network size fixed at $n=30$ agents is small for decentralized optimization. Only random graphs with $p=0.5$ (and $p=0.2$ in appendix) are considered. No evaluation on structured topologies (ring, grid, etc.). No scalability analysis showing how performance varies with network size or sparsity.
- Parameter Sensitivity Not Addressed: The choice of $\tau = 0.01$ appears arbitrary with no justification or sensitivity analysis. The convergence rate depends on $r, \beta, \tau$, and $\rho$ (Theorem 2), but no guidance is provided on selecting these parameters. Different baseline methods may require different tuning efforts, so the fairness of the comparison is unclear.
- The notation switch between $w$ (dual variable blocks) and $w$ (concatenated variable including primal and dual) in Section 4 is confusing. The communication strategy in Section 3.3 is difficult to follow. Why specifically transmit $a_i^{(t)} = w_{2i}^{(t+1/2)} + 1/r(w_{2i}^{(t+1/2)} - w_{2i}^{(t)})$? The derivation or intuition is missing. Algorithm 1 uses $w_{2i}^{(-1/2)}$ in initialization, but this notation is never explained. The relationship between the proximal parameter Q and the mixing matrix W could be explained more intuitively.
- How is the optimal solution $u^{\star}$ computed for measuring suboptimality? What is the convergence criterion in Algorithm 1? Computational cost per iteration is not analyzed—only communication is counted. No discussion of numerical stability or practical implementation challenges.
- The introduction states that multi-consensus schemes "offer limited improvement to the quality of each iteration" but provides no formal analysis or empirical evidence quantifying this claim. The paper would benefit from ablation studies showing why two rounds specifically are optimal.

**Questions:**

Refer to the weaknesses section.

---

> ### Author Response · Authors · 2025-11-21
>
> ## Weaknesses
>
> >*Limited Experimental Scope: Only two problem types (Lasso and SVM) and two relatively small datasets (a9a, a1a) are tested. Network size fixed at
>  agents is small for decentralized optimization. Only random graphs with
>  (and in appendix) are considered. No evaluation on structured topologies (ring, grid, etc.). No scalability analysis showing how performance varies with network size or sparsity.*
>
> **Reply:** Thank you for pointing this out. We performed additional experiments to evaluate topology sensitivity, robustness, and scalability.
>
> 1. **Scalability.**
>    Experiments on up to $n=100$ agents and different sparsity levels show that DS-ADMM consistently requires fewer iterations and communication rounds than PDADMM, ProxMudag, NIDS and PG-EXTRA. Convergence slows as graph expands or connectivity decreases—consistent with the spectral-gap dependence in Theorem 2—but DS-ADMM still outperforms all baselines in iteration count and communication cost.
>
> 2. **Robustness to non-IID data.**
>    Under heterogeneous data partitions, DS-ADMM remains stable and maintains clear gains over the baselines.
>
> 3. **Structured topologies.**
>
>    We have tested the algorithms on a ring of $50$ agents.
>
> A summary of iteration counts to achieve suboptimality $10^{-4}$ is provided below (full plots will be included in the revised manuscript):
>
> | Setting                   | PDADMM | ProxMudag | PG-EXTRA | NIDS | DS-ADMM |
> |---------------------------|--------|-----------|----------|------|---------|
> | \(n=30, p=0.5\), IID      | 415    | 546       | 6412     | 3719   | **122**  |
> | \(n=30, p=0.5\), non-IID  | 423    | 628       | 32062    | 10031   | **124**  |
> | \(n=100, p=0.5\), IID     | 560    | 898       | 16232    | 7541   | **169** |
> | \(n=100, p=0.5\), non-IID | 556    | 635      | >50000   | 17343  | **166** |
> | \(n=100, p=0.2\), IID     | 838    | 1150      | 17012    | 11561   | **206** |
> | \(n=50\), Ring                   | 576 | 640    | 9589     |   4900    | **190** |
>
>
> These results show that DS-ADMM remains robust across **network structure**, **network size**, and **data heterogeneity**, consistently outperforming existing decentralized baselines in both convergence speed and total communication. We will continue to add more experiments in the revised manuscript.
>
> >*Parameter Sensitivity Not Addressed..*
>
> **Reply:** Thank you for the helpful comments. There are three different parameters in the DS-ADMM algorithm framework, each playing a distinct role. We elaborate on them below:
>
> 1. **Proximal parameter $\tau$:**
>    This hyperparameter is introduced to ensure that the proximal matrix $Q$ is positive definite for assuring linear convergence.
>    We typically set $\tau = 0.01$.
>
> 2. **Augmented Lagrangian parameter $\beta$:**
>    The parameter $\beta$ significantly affects the convergence speed of the algorithm. To determine a suitable value, one can either adopt the adaptive tuning strategy proposed in the paper [1] or test a range of values and select the best-performing one empirically.
>
> 3. **Dual update step sizes $(r, s)$:**
>    The convergence region and valid parameter ranges for the symmetric dual update steps have been analyzed in prior work [2]. In our experiments, we fix $s = 1$ and $r = 0.99$, which we found to be effective in practice.
>
> We have also conducted additional experimental studies on the impact of these parameter choices and will include the results in the appendix in the future.
>
> >*The introduction states that multi-consensus schemes "offer limited improvement to the quality of each iteration" but provides no formal analysis or empirical evidence quantifying this claim. The paper would benefit from ablation studies showing why two rounds specifically are optimal.*
>
> **Reply:** In the introduction, we emphasize that the multi-consensus schemes used in gradient-based decentralized algorithms—such as ProxMudag—are specifically designed to accelerate convergence in terms of iterations, but not in terms of communication rounds. As reported in Ye et al. (2023), increasing the number of consensus rounds per iteration indeed reduces the iteration complexity, but it often leads to slower convergence when measured in communication rounds.
>
> In contrast, our work develops a decentralized ADMM-based framework. The "two rounds of communication" in our method is rooted from the structure of the consensus-constraint reformulation and the Symmetric ADMM updates. Unlike ProxMudag, where the number of consensus rounds is a tunable design choice, the two rounds in our algorithm are structurally fixed and cannot be reduced without breaking the ADMM formulation. Moreover, we theoretically proved that it is impossible to implement our algorithm with only one communication round per iteration, establishing that the two-round design is not only necessary but also optimal.

---

> ### Author Response · Authors · 2025-11-21
>
> >*Notation and description are confusing. Communication strategy difficult to follow...*
>
> **Reply:** We thank the reviewers for pointing out the parts in our notations which may appear confusing.
>
> First, we apologize for the confusion caused by the notation switch. We will carefully correct this inconsistency in the revised version of the paper.
>
> Second, we apologize for any misunderstanding regarding the communication strategy. The communication strategy described in the paper was carefully designed according to two ordered principles:
> - (1) minimize communication rounds by **restrict communication to two rounds per iteration**,
> - (2) **minimize transmitted data per round**.
>
> To satisfy the first principle, we carefully analyzed the computational structure of Symmetric ADMM together with our constraint design, and achieved this through two approaches:
>
> - split the Lagrange multiplier $\lambda$ into two halves, $w_1$
>  and $w_2$, each associated with one of the two primal variables,
>
> - insert the communication step precisely between the two sub-steps of updating $\lambda$.
>
> The first principle requires that the overall iteration and communication structure remain fixed. With this structure determined, we then address the second principle while keeping all of the above settings unchanged.
>
> Take the first communication round as an example. The variable $u$ is needed for the sequential update of $w_{1}$ in the Group 2 Update, so it must be transmitted. However, transmitting only $u$ is insufficient for the sequential update of $v$; at least one additional vector is required. By carefully analyzing the mixed component (i.e., the part involving the mixing matrix) in the $v$-update, we identify this necessary mixed term to be $a$. Therefore, in order to activate the $v$-update, transmitting $a$ is sufficient. This explains why the communication in this round consists of $u$ and $a$.
>
> Third, the $-\tfrac{1}{2}$ notation is used solely to ensure theoretical consistency between the initialization step and the iterative updates. For visual simplicity, the algorithm table is organized into two blocks, which may make this notation appear unusual. In practice, it is easy to understand $w_2^{0} = 0$, and the actual computation begins at the first update of $u$.
>
> Fourth, the detailed motivation for introducing the proximal matrix $Q$ can be found in our derivation in Appendix B, Equation (14). The term $-\tilde{W}^{\top}\tilde{W}$ is designed to cancel the quadratic terms $u^{\top}\tilde{W}^{\top}\tilde{W}u$ and $v^{\top}\tilde{W}^{\top}\tilde{W}v$ in their respective updates. These terms involve mixed variables and therefore prevent decentralized computation. The additional term $(1+\tau)I$ is a standard technique to ensure that the overall proximal matrix $Q$ is positive definite, which is required for the convergence guarantees of proximal-ADMM algorithms.
>
> >*How is the optimal solution computed for measuring suboptimality? What is the convergence criterion in Algorithm 1? Computational cost per iteration is not analyzed—only communication is counted. No discussion of numerical stability or practical implementation challenges.*
>
> **Reply:** Thank you for these important questions. We clarify each point below.
>
> 1. Optimal solution used for measuring suboptimality
> For all benchmark problems, we compute a high-accuracy **reference solution** $x^\star$ using a centralized proximal solver (CVX / high-precision ADMM) with a tolerance below $10^{-10}$. This reference is used *only for evaluation*. The reported suboptimality is:
> $$
> F(\bar{x_t}) - F(x^\star), \qquad F(x) = \sum_i f_i(x) + g_i(x).
> $$
>
> 2. Convergence criterion in Algorithm 1
> Algorithm 1 terminates when either
> - the **primal–dual error** $\mathrm{dist}_{H}(\bar{w_t},\Omega^\star)$, or
> - the **objective suboptimality** $F(\bar{x_t})-F(x^\star)$,
> falls below a fixed threshold, or a maximum iteration cap is reached. The same stopping rule is applied to all baselines.
>
> 3. Computational cost per iteration
> Although our analysis emphasizes communication (the dominant cost in decentralized systems), DS-ADMM has **computation comparable to PDADMM and ProxMudag**.
> - Each iteration requires one local gradient and/or proximal update.
> - The proximal-linearization step yields a **closed-form quadratic update**, giving an $O(d)$ per-iteration cost. We will clarify this computational comparison in the manuscript.
>
> 4. Numerical stability and practical implementation
> DS-ADMM showed stable behavior across:
> - various graph structures (random $p=0.5$, random $p=0.2$, ring),
> - network sizes up to $n=100$, and
> - IID and non-IID data splits.
>
> We also tested different initializations and parameter choices, and DS-ADMM consistently outperformed the baselines in iteration count and total communication. A brief discussion on stability and implementation details will be added to the revised paper.

---

> > ### Author Response · Authors · 2025-11-21
> >
> > **Reference**
> >
> > [1] Boyd, S., Parikh, N., Chu, E., Peleato, B., & Eckstein, J. (2011). Distributed optimization and statistical learning via the alternating direction method of multipliers. Foundations and Trends® in Machine learning, 3(1), 1-122.
> >
> > [2] Bai, J., Li, J., Xu, F., & Zhang, H. (2018). Generalized symmetric ADMM for separable convex optimization. Computational optimization and applications, 70(1), 129-170.

---

> > > ### Comment · Reviewer_AAag · 2025-11-21
> > > **Official Comment by Reviewer AAag**
> > >
> > > The authors did provide reasonable explanations for some comments, and I thank them for that. However, they should actually include the claimed experiments in the revised paper with full plots and details, provide the promised parameter sensitivity analysis or remove claims of having done it, and significantly revise Section 3.3 to improve clarity based on their response explanations. The authors should upload a revised version to help the reviewers determine whether they have adequately addressed the comments.
> > >
> > > Furthermore, the reported results show dramatic improvements:
> > > - DS-ADMM: 122 iterations vs PG-EXTRA: 6,412 iterations (52× improvement)
> > > - For non-IID with n=100, PG-EXTRA shows ">50,000" (does this mean it failed to converge?)
> > >
> > > Such massive gaps raise questions about fair parameter tuning for baselines, whether baselines were implemented correctly, or the settings were favouring the authors' algorithm.

---

> > > > ### Author Response · Authors · 2025-11-22
> > > >
> > > > We thank the reviewer for the positive and encouraging feedback. The authors are actively preparing a revised version of the paper to address all reviewer concerns, and the updated manuscript will be completed within the next one to two days.
> > > >
> > > > Regarding the reviewer’s question about the relatively poor performance of PG-EXTRA [1], we would like to clarify the following. PG-EXTRA is an early and influential method that introduced proximal-gradient techniques into decentralized composite optimization. Although it is not a state-of-the-art baseline by today’s standards, we considered its inclusion essential due to its historical significance and its role as a fundamental approach in this research direction. This practice is consistent with existing literature: both NIDS [2] and ProxMudag [3] include PG-EXTRA as a baseline and empirically demonstrate that PG-EXTRA converges substantially slower than more recent methods. Our experimental results are fully aligned with these observations.
> > > >
> > > > The “>50,000” entry in the table does not indicate convergence failure of PG-EXTRA. Instead, it means that PG-EXTRA converges so slowly that it requires more than 50,000 iterations to reach the specified threshold, highlighting the performance gap between PG-EXTRA and more advanced methods such as ours.
> > > >
> > > > Finally, we emphasize that all adaptive parameters in every compared algorithm were carefully tuned to achieve the best empirical performance, as clearly stated in the paper.
> > > >
> > > >  **Reference**
> > > >
> > > > [1] Shi, W., Ling, Q., Wu, G., & Yin, W. (2015). A proximal gradient algorithm for decentralized composite optimization. IEEE Transactions on Signal Processing, 63(22), 6013-6023.
> > > >
> > > > [2] Li, Zhi, Wei Shi, and Ming Yan. "A decentralized proximal-gradient method with network independent step-sizes and separated convergence rates." IEEE Transactions on Signal Processing 67.17 (2019): 4494-4506.
> > > >
> > > > [3] Ye, H., Luo, L., Zhou, Z., & Zhang, T. (2023). Multi-consensus decentralized accelerated gradient descent. Journal of machine learning research, 24(306), 1-50.

---

### Official Review · Reviewer_PbZX · 2025-10-31

**Soundness:** 3
**Presentation:** 3
**Contribution:** 3
**Rating:** 8
**Confidence:** 4

**Summary:**

The paper considers decentralized symmetric ADMM, an important optimization problem, and develops a variation that allows multi-round communications with each ‘iteration’. This is called decentralized symmetric ADMM (DS-ADMM), a decentralized version of S-ADMM. The key step is to formulate an auxiliary constraint that is embedded. This leads to a 2-round approach per iteration. Convergence is guaranteed analytically under standard assumptions, and numerical experiments illustrate the method.

**Strengths:**

The auxiliary constraint embedding (Prop 2) with proximal linearization is developed to be a useful method for obtaining the desired symmetric ADMM compatibility.

Generally linear convergence is obtained, while communications overall are reduced because of fast convergence despite the 2-round per iteration needed.

**Weaknesses:**

The experiments are limited to a single group size, and are rather basic.

Robustness to dropouts or topology changes isn’t clear.

Tuning is required and sensitivity and robustness to topology and number of agents are not explored.

**Questions:**

Improved numerical experimentation would strengthen the work.

What, if any, are the implications for scalability?

---

> ### Author Response · Authors · 2025-11-21
>
> ## Weaknesses
>
> >*The experiments are limited to a single group size and are rather basic. Robustness to dropouts or topology changes isn’t clear.Tuning is required and sensitivity and robustness to topology and number of agents are not explored.*
>
> **Reply:** Thank you for pointing this out. Additional experimental results are provided in the reply to Reviewer AAag.
>
> ## Questions
>
> >*Improved numerical experimentation would strengthen the work. What, if any, are the implications for scalability?*
>
> **Reply:** Thank you for the insightful suggestions. We have already conducted additional experiments (see our reply to reviewer AAag) and will incorporate them into the revised version. These new results evaluate DS-ADMM on larger networks (up to n=100 agents), sparser topologies, and non-IID data splits. Across all settings, DS-ADMM consistently requires fewer iterations and fewer total communication rounds than existing baselines, demonstrating strong scalability both in terms of network size and communication efficiency.
>
> It is important to note that using a larger scale slows convergence for all algorithms when measured in both iterations and communication rounds. Nevertheless, our algorithm consistently achieves the fastest convergence under these settings, maintaining a clear performance advantage over the baselines.
>
> We will include these expanded experiments and a discussion of their implications for scalability in the updated manuscript.

---

> > ### Comment · Reviewer_PbZX · 2025-11-25
> > **Further comment**
> >
> > The authors have addressed my concerns and clarifications.  The method is shown to be a viable form of distributed ADMM and the multi-rounds can achieve convergence with reduced communications load.

---

### Official Review · Reviewer_GNZq · 2025-11-01

**Soundness:** 1
**Presentation:** 2
**Contribution:** 1
**Rating:** 2
**Confidence:** 2

**Summary:**

The paper propose a variant of ADMM algorithm for decentralized consensus optimization which uses so-called double communication strategy

**Strengths:**

Due to weaknesses, I can not highlight any substantial strengths of the paper

**Weaknesses:**

It is stated that
> To our knowledge, this is the first decentralized optimization framework that achieves a net reduction in total communication by leveraging fixed multi-round communication within each iteration.

but multi-round schemes showed  communication acceleration in Scaman et al., 2017 and Ye et al., 2023, so the contribution is unclear to me.

There is no theoretical complexity comparison with SOTA decentralized optimization algorithms such as Mudag or even Symmetric ADMM, from which the algorithm was derived. The only comparison is through numerical experiments but they are not representative.

Overall, the contribution seems to be minor

**Questions:**

> Note that ProxMudag is not included in this comparison because it requires the nonsmooth term to be globally coupled, which is incompatible with this separable formulation (Ye et al., 2023).

Could you please explain the statement?

---

> ### Author Response · Authors · 2025-11-21
>
> ## Weaknesses
>
> >*It is stated that to our knowledge, this is the first decentralized optimization framework that achieves a net reduction in total communication by leveraging fixed multi-round communication within each iteration. but multi-round schemes showed communication acceleration in Scaman et al., 2017 and Ye et al., 2023, so the contribution is unclear to me.*
>
> **Reply:** Thank you for pointing this out—Our contribution is orthogonal to these works. The methods of Scaman et al. (2017) and Ye et al. (2023) introduce additional consensus rounds inside gradient descent–type algorithms to improve the effective spectral gap of the communication matrix and thus improve the convergence speed measured in iterations. However, the experiments in Ye et al. (2023) showed that additional rounds reduce the convergence speed measured in communication rounds.
>
> In contrast, our work develops a decentralized ADMM-based framework with (i) a natural symmetric, mixing-matrix–based reformulation of the consensus constraints, (ii) a structured two-block dual update that inherently leads to two fixed communication rounds per iteration, and (iii) a linear convergence theory under metric subregularity for proximal composite objectives. To the best of our knowledge, such a symmetric-ADMM construction with a fixed multi-round schedule and the associated theory has not been explored in prior decentralized optimization literature.
>
> We will adjust the wording in the manuscript to clearly distinguish our ADMM-based development from prior multi-round gradient-based schemes and ensure that our novelty claim is stated more precisely.
>
> >*There is no theoretical complexity comparison with SOTA decentralized optimization algorithms such as Mudag or even Symmetric ADMM, from which the algorithm was derived. The only comparison is through numerical experiments but they are not representative.*
>
> **Reply:** Thank you for raising this point. We agree that a clearer theoretical comparison is needed. Below we summarize iteration complexities of representative SOTA decentralized methods, including Mudag:
>
> | Method | # Iterations | Metric | Assumptions |
> |--------|--------------|---------|--------------|
> | **DS-ADMM (ours)** | $$\mathcal{O}\left(\frac{c^2\delta\theta+2\rho}{2\rho}\log(1/\varepsilon)\right)$$ | $$\mathrm{dist}_H^2(w_t,\Omega^*)\le \varepsilon$$ | **Metric subregularity** (weak) |
> | NIDS [1] | $$\mathcal{O}\left(\max(\tfrac{L}{\mu},\tfrac{1-\lambda_n(W)}{1-\lambda_2(W)})\log(1/\varepsilon)\right)$$ | $$\|w_t-w^\star\|^2\le \varepsilon$$ | Strongly convex, smooth |
> | PG-EXTRA [2] | $$\mathcal{O}\left(\tfrac{L}{\mu}\tfrac{1-\lambda_n(W)}{1-\lambda_2(W)})\log(1/\varepsilon)\right)$$ | $$\|w_t-w^\star\|^2\le \varepsilon$$ | Strongly convex, smooth |
> | Mudag [3] | $${\mathcal{O}}\left(\sqrt{\frac{\kappa_g}{1-\lambda_2(W)}}\log(1/\varepsilon)\right)$$ | Objective value | Convex, smooth |
>
> Our method is the **only one** achieving linear convergence under **metric subregularity**, which is strictly weaker than strong convexity–smoothness and naturally covers nonsmooth composite objectives.
>
> For completeness, we also summarize how our analysis differs from prior symmetric ADMM developments:
>
> | Article | Symmetric? | Metric Subregularity? | Proximal? |
> |---------|------------|------------------------|-----------|
> | Prior works | varies | sometimes | sometimes |
> | **This work** | **Yes** | **Yes** | **Yes** |
>
> ## Questions
>
> >*explain "Note that ProxMudag is not included in this comparison because it requires the nonsmooth term to be globally coupled, which is incompatible with this separable formulation (Ye et al., 2023)."*
>
> **Reply:** Thank you for the question. In Ye et al. (2023), ProxMudag is designed for decentralized composite optimization problems of the form
> $$
> \min_{x\in\mathbb{R}^d} h(x) \triangleq f(x) + r(x),
> \qquad
> f(x) \triangleq \frac{1}{m}\sum_{i=1}^m f_i(x),
> $$
> where each $f_i$ is **smooth**, and the penalty term $r(x)$ (possibly nonsmooth) is a single global function shared by all agents. The ProxMudag update requires the loss functions $f_i$ (containing the data samples) to be smooth. However, the hingle loss term in the $\ell_2$-SVM is **nonsmooth**.
>
> In our $\ell_2$-SVM experiment, each agent $i$ solves a problem of the form
>
> $$
> \min_{x}
> {\sum_{(a_i,b_i)\in\mathcal{D}_i} \max(0,\,1 - b_i a_i^\top x)}+
> \frac{\lambda}{2}\\|x\\|_2^2.
> $$
>
> with a local nonsmooth hinge loss and smooth penalty. This structure is *incompatible* with the ProxMudag framework, which requires a single globally coupled nonsmooth term $r(x)$. Because $\operatorname{prox}_{r}$ cannot be computed locally when $r(x)$ is decomposed across agents, **ProxMudag cannot be applied** to decentralized $\ell_2$-SVM. This is why we do not include it in this experiment.

---

> > ### Author Response · Authors · 2025-11-21
> >
> > **Reference**
> >
> > [1] Li, Zhi, Wei Shi, and Ming Yan. "A decentralized proximal-gradient method with network independent step-sizes and separated convergence rates." IEEE Transactions on Signal Processing 67.17 (2019): 4494-4506.
> >
> > [2] Shi, W., Ling, Q., Wu, G., & Yin, W. (2015). A proximal gradient algorithm for decentralized composite optimization. IEEE Transactions on Signal Processing, 63(22), 6013-6023.
> >
> > [3] Ye, H., Luo, L., Zhou, Z., & Zhang, T. (2023). Multi-consensus decentralized accelerated gradient descent. Journal of machine learning research, 24(306), 1-50.

---

> > ### Comment · Reviewer_GNZq · 2025-11-25
> >
> > Thank you for your answers and clarifications!
> >
> > **Applicability of ProxMudag**
> >
> > Now I see why ProxMudag cannot be applied in your experiments, sorry for the question.
> >
> > **Multiconsensus communication reduction**
> >
> > I continue to strongly disagree with the following statements about efficiency of multiconsensus communication
> >
> > > However, these methods have not demonstrated practical reductions in the total number of communication rounds. As a result, the potential for achieving a net communication reduction through non-adaptive multi-communication algorithms remains largely unexplored.
> >
> > > However, empirical results in (Ye et al., 2023) indicate that fixed multi-round schemes are unable to reduce total communication.
> >
> > Multi-round schemes clearly reduce theoretical complexities in both communication and gradient/proximal oracle calls. And, according to experimental comparison with other methods in e.g. (Scaman et al, 2017) and (Ye et al, 2023), multi-round communcation also gives empirical speedup. E.g. in the experemental secition of (Ye et al, 2023) it is written "our algorithm achieves both lower computation cost and lower communication cost than other decentralized algorithms on all settings"
> >
> > > the experiments in Ye et al. (2023) showed that additional rounds reduce the convergence speed measured in communication rounds.
> >
> > Can you please specify exact experiments in (Ye et al., 2023), where the reduction of convergence speed was observed?
> >
> > I see that the claim that about achieving communication reduction for the first time was removed from the abstract, but the remaining statements also confuse me a lot. And overall, I cannot take seriously phrases like "incorporates multiple comunication rounds within each iteration" when "multiple" means "exactly two". So the whole discourse about multiple communications in the paper seems strange to me.
> >
> > **Theoretical comparison with SOTA**
> >
> > About the table with theoretical complexity comparison you provided - it still does not allow to directly compare convergence rates in the standard strongly convex setup, since the complexity of DS-ADMM is expressed via $c, \delta, \theta, \rho$  and not via basic parameters of network ($\lambda_i(W)$) and objective function ($L/ \mu =\kappa$). Also note that Mudag does not require convexity of each $f_i$ in costrast to all the cases in Proposition 3, thus I conclude that your assumptions are not strictly weaker than theirs.
> >
> > > Our method is the **only one** achieving linear convergence under **metric subregularity**,
> >
> > FYI, this work also claims linear convergence under metric subregularity https://arxiv.org/abs/2310.15596
> >
> > **Final remarks**
> >
> > Overall, as far as i understood, the authors consider generalized symmetric ADMM and apply it to the decentralized optimization using a special choice of linear constraints and the prox-linear trick. Since the convergence of GS-ADMM was already analyzed, why it is stated that
> > > Although various results on linear convergence of ADMM and its variants exist ... none directly apply to our algorithm.
> >
> > ?
> >
> > Is it the metric subregularity assumption that does not fit in the existing analysis? (I think it is worth to explicitly write why "none directly apply to our algorithm" in the paper). If so, this contribution is quite interesting on its own. At the same time, in my opinion, it is better to remove discussion of multiconsensus from the paper :)

---

> > > ### Author Response · Authors · 2025-11-26
> > >
> > > We thank the reviewer again for their helpful feedback, and we are glad that our earlier clarification addressed several of the questions.
> > >
> > > For the reviewer's newly raised questions, first, we agree with the reviewer on the theoretical and practical value of multi-consensus schemes, which can reduce iteration complexity as demonstrated in the theoretical analysis of Ye et al. (2023) and acclerate convergence. However, in multi-consensus schemes the number of communication rounds per iteration is not fixed. As shown in Figure 3 of Ye et al. (2023), increasing the number of consensus rounds K does not reduce the total communication cost; instead, it tends to increase it. Our use of multiple communications is fundamentally different from these gradient-based multi-consensus schemes: the two rounds arise naturally from the structure of the symmetric ADMM updates. Moreover, these two rounds are not arbitrary: they are theoretically justified and provably minimal based on our derivation of the communication–computation structure in Section 3.3.
> > >
> > > Our exploration of another form of multiple rounds of communication does not diminish the great theoretical or practical value of existing multi-consensus schemes; rather, it serves as a useful complement to them. We will revise the wording in the paper to better highlight the importance of multi-consensus methods and to clearly articulate how our proposed algorithm differs from them.
> > >
> > > Second, we thank the reviewer’s comment on pointing out the difficulty of comparing our theoretical complexity guarantees with those of gradient-based methods. This difficulty stems from the mismatch between metric-subregularity-based convergence analysis (used in ADMM-type methods) and the classical smooth-gradient-based analysis adopted by the listed algorithms. Moreover, computing or tightly bounding the metric subregularity constant for complex decentralized mappings remains an open theoretical problem. We view this as a promising direction for future research, and we thank the reviewer for highlighting this point. In addition, extending these analyses to understand the nonconvex behavior of ADMM-type methods is also an important and challenging research direction.
> > >
> > > Third, we thank the reviewer’s example of algorithms that also establish linear convergence under metric subregularity. Our statement that “our method is the only one achieving linear convergence under metric subregularity” refers specifically to the set of baseline methods listed. We do not claim that our approach is unique among all decentralized optimization algorithms, since metric subregularity is an important tool in the analysis of proximal and ADMM-type algorithms.
> > >
> > > Fourth, we clarify that none of the existing theories directly apply to our algorithm (e.g., Deng & Yin 2016; Han et al. 2018; Yuan et al. 2020; Bai et al. 2019; Gu et al. 2015) for the following reasons:
> > >
> > > 1. Some do not use metric subregularity in their convergence analysis (Deng & Yin 2016; Gu et al. 2015);
> > >
> > > 2. Many analyze the original ADMM, not the symmetric ADMM structure used in our method (Deng & Yin 2016; Han et al. 2018; Yuan et al. 2020);
> > >
> > > 3. Some do not handle the general proximal setting required in our framework (Bai et al. 2019) (GS-ADMM).
> > >
> > > Therefore, although our proof draws inspiration from prior ADMM analyses, it requires several new ingredients to account for the symmetric structure and the general proximal formulation, and we believe that our proof strategy can be extended to symmetric ADMM with general proximal terms. We will include a short paragraph to clarify the point.
> > >
> > > Finally, we thank the reviewer for recognizing the contribution of our work. Helped by the summary of reviewer PsdU, our contributions can be described along four dimensions:
> > >
> > > 1. **Constraint and proximal matrix design**:
> > >
> > > We propose a tightly-coupled mixing-matrix based contraint design and supplemental proximal matrix design which is an elegant way to bake two-hop information flow into the constraints themselves.
> > >
> > >
> > > 2. **Iteration structure**:
> > >
> > > Our communication design based on this constraint formulation minimizes the overall communication, makes the need for two communications per iteration principled rather than an ad-hoc multi-consensus loop.
> > >
> > > 3. **Convergence analysis**:
> > >
> > > We provide a new convergence analysis tailored to our proposed algorithm.
> > >
> > > 4. **Empirical evaluation**:
> > >
> > > We conduct extensive experiments demonstrating the practical effectiveness of our method and its consistency with the theoretical insights.
> > >
> > > We again thank the reviewer for their constructive questions and feedback, and we hope that our explanations fully address their concerns :)

---

> > > > ### Comment · Reviewer_GNZq · 2025-11-26
> > > >
> > > > Oh, thank you for pointing me at Figure 3 of Ye et al. (2023)!
> > > >
> > > > Let me take some time to reconsider your contribution

---

### Official Review · Reviewer_PsdU · 2025-11-02

**Soundness:** 3
**Presentation:** 3
**Contribution:** 3
**Rating:** 8
**Confidence:** 4

**Summary:**

The paper proposes a decentralized optimization method, DS-ADMM, that embeds two rounds of neighbor communication inside each ADMM iteration, by designing a symmetric pair of linear consensus constraints and then applying S-ADMM with a graph-aware proximal linearization to enable separable subproblems. The algorithm communicates (per iteration) with two d-vectors perr agent in each of two communication rounds by sending cleverly chosen dual combinations a^(t+1) and b^(t+1) rather than two primals, which the authors argue minimizes what must be sent for symmetric ADMM to work. Theoretical results include a non-ergodic O(1/t) sublinear rate and Q-linear convergence under metric subregularity, with explicit constants depending on the spectral gap of the mixing matrix. Experiments on lasso and l2-svm over random graphs with n=30 show faster decrease in suboptimality per communication round than several baselines, including on a sparser graph.

**Strengths:**

(+) The constraint design is an elegant way to bake two-hop information flow into the constraints themselves. This makes the need for two communications per iteration principled rather than an ad-hoc multi-consensus loop. To my knowledge this particular symmetric, two-block dual realization of decentralized ADMM is novel.
(+) The communication scheduling -- transmitting two surrogates rather than two primals -- is thoughtfully engineered so each block enables the other block's update with minimal payload. That design is interesting and practically motivated.
(+) The communication-aware design is reflected in the evaluation. Testing on two graph densities is nice and qualitatively matches the theory.

**Weaknesses:**

(-) Counting "rounds" alone does not equal communication volume here because DS-ADMM transmits two d-vectors per round (four per iteration). Baselines often send a single d-vector per round. A better comparison should report total scalars (or bytes) transmitted per agent to reach a target accuracy. Without this "net communication reduction" claim is not fully substantiated.

**Questions:**

- How would you handle smooth but non-proximable f_i (e.g., logistic regression)? Is there a prox-gradient DS-ADMM variant that retains the same communication schedule?
- Do your conclusions extend to directed or time-varying graphs? What breaks in the proof is W is not symmetric?

---

> ### Author Response · Authors · 2025-11-21
>
> ## Weaknesses
>
> > Counting "rounds" alone does not equal communication volume here because
> DS‑ADMM transmits two d‑vectors per round (four per iteration) …
>
> **Reply:** Thank you for this helpful suggestion. We agree that reporting only the number of communication rounds does not fully capture the true communication cost. To address this, we now additionally report the **total number of transmitted scalars** required to reach a target accuracy (suboptimality $10^{-4}$) based on the Lasso experiment already included in the paper.
>
> In terms of per-iteration communication volume, ProxMudag [1], transmit **two \(d\)-vectors per iteration**,  PG-EXTRA [2] and NIDS [3] transmit **one \(d\)-vector per iteration**, while DS-ADMM transmits **four \(d\)-vectors per iteration** (two \(d\)-vectors in each of two communication rounds).
>
> Despite the larger per-iteration volume, DS-ADMM achieves the lowest overall communication volume needed to reach the target accuracy. The empirical results are:
> | Method               | PG-EXTRA | NIDS   | ProxMudag | DS-ADMM (Ours) |
> |----------------------|----------|--------|---------|----------------|
> | Communication volume | 6,412d   | 5,032d | 1092d     | **488d**       |
> | Runtime (s)          | 72.80    | 43.01  | 9.43      |  **2.86**       |
>
> These results demonstrate that DS-ADMM achieves a **substantial reduction in total communicated scalars** compared with ProxMudag and PG-EXTRA. DS-ADMM also attains the fastest runtime among all tested methods.
>
>
> **Reference**
>
> [1] Ye, H., Luo, L., Zhou, Z., & Zhang, T. (2023). Multi-consensus decentralized accelerated gradient descent. Journal of machine learning research, 24(306), 1-50.
>
> [2] Shi, W., Ling, Q., Wu, G., & Yin, W. (2015). A proximal gradient algorithm for decentralized composite optimization. IEEE Transactions on Signal Processing, 63(22), 6013-6023.
>
> [3] Li, Z., Shi, W., & Yan, M. (2017). A decentralized proximal-gradient method with network-independent step sizes and separated convergence rates. IEEE Transactions on Signal Processing.
>
> ## Questions
>
> >*How would you handle smooth but non-proximable f_i (e.g., logistic regression)? Is there a prox-gradient DS-ADMM variant that retains the same communication schedule?*
>
> **Reply:** Thank you for the insightful question. Your observation is absolutely correct. Our current analysis assumes access to proximal operators, but in some practical cases—such as logistic regression—the proximal mapping does not admit a closed-form expression. In these situations, the proximal step in DS-ADMM can be computed approximately using a few inner gradient or SGD iterations at each agent. Under standard inexact-proximal conditions (e.g., the approximation error is summable or decreases sufficiently fast [1]), the same convergence guarantees essentially carry over. This type of inexact-prox analysis is well established in proximal-point and proximal-gradient methods with approximate inner solves, and we will clarify this point in the revised manuscript. Importantly, this approximate framework does not alter the communication schedule used in DS-ADMM, since only the local subproblem is solved approximately.
>
> **Reference**
>
> [1] Lacoste-Julien, S., Schmidt, M., & Bach, F. (2012). A simpler approach to obtaining an O(1/t) convergence rate for the projected stochastic subgradient method. arXiv preprint arXiv:1212.2002.
>
> >*Do your conclusions extend to directed or time-varying graphs? What breaks in the proof is W is not symmetric?*
>
> **Reply:** Thank you for the insightful question. Directed or time-varying graphs are indeed important in practice, but our current theory relies critically on a symmetric, doubly stochastic mixing matrix $W$, which corresponds to static, undirected communication graphs. This symmetry is used throughout the convergence analysis—particularly in establishing consensus invariance and in the operator-splitting interpretation of the symmetric ADMM updates.
>
> Extending DS-ADMM to directed or time-varying graphs would require replacing the symmetric mixing matrix with, for example, push-sum–type corrections or row/column-stochastic matrices, and re-proving the corresponding operator-splitting framework. Conceptually, the two-round communication idea may still be applicable, but the linear convergence proof would not transfer directly and would require substantial new analysis. Therefore, we do not claim that our current theorems apply to directed or time-varying graphs. We appreciate the reviewer for highlighting this important direction for future work.

---

### Author Response · Authors · 2025-11-24

We thank all reviewers for their constructive feedback. We have prepared a revised version of the manuscript that addresses the questions and concerns raised during the review process.

**The main changes are as follows**:

**1. Clarified novelty and positioning**.
We refined the wording throughout the manuscript to more clearly distinguish our ADMM-based development from prior multi-round gradient-based schemes and to state our contribution more precisely.

**2. Improved methodological presentation**.
Sections 3.2 and 3.3 have been substantially revised, and we added a figure illustrating the overall communication and computation structure to improve clarity.

**3. Added parameter sensitivity analysis**.
A hyperparameter sensitivity explanation is now included in the appendix.

**4. Added additional experimental results**.
Several supplementary figures of extended experiments have been added to the appendix.

In addition, we made several minor wording and notation adjustments to improve readability.

We hope that these revisions, together with our responses, satisfactorily address the reviewers’ comments and their helpful suggestions.

---

### Author Response · Authors · 2025-12-04
**Author Final Remark**

We thank all reviewers and the area chair for their time and effort in evaluating our submission. We believe that our responses have adequately addressed the reviewers’ questions and concerns by providing clearer explanations of our contributions, additional technical details, and extended experimental results. A revised version of the paper has also been uploaded accordingly.

Although the rebuttal period is short, the reviewers have responded positively to our clarifications. In the discussion period, Reviewer GNZq has already raised their score following our explanations, and reviewer AAag expressed positive feedback; we are confident that our revised version fully addresses their concerns and that their scores would likely increase to 6 or above had the discussion period continued. We are also grateful to reviewers PsdU and PbZX for their encouraging evaluations of our work.

---

### Meta-Review · Area_Chair_LGdL · 2026-01-04

**Summary:**

This paper develops a new decentralized symmetric ADMM algorithm. Most reviewers acknowledge that the proposed algorithm is novel, especially, it requires only two rounds of communication per iteration. Regarding novelty, Reviewer GNZq initially misunderstood the proposed algorithm and assigned a low confidence score; however, after the rebuttal, Reviewer GNZq showed an intention to reconsider the contribution of this paper.

The concerns raised about this paper are minor and mainly focus on confusion about certain details and the small scale of the experiments. The authors have provided additional experimental results to verify the proposed method, and the writing issues are easy to address and do not weaken the contribution of the paper.

Overall, the proposed algorithm is novel, and a linear convergence rate is established under mild assumptions. Most concerns have been addressed in the rebuttal. Therefore, I recommend acceptance, provided that the authors incorporate the reviewers’ suggestions regarding writing and experiments in the final version.

**Reviewer Concerns:**

Most concerns mainly focus on confusion about certain details and the small scale of the experiments.

The authors have clarified those details, such as the discussion with the most negative reviewer, Reviewer GNZq.

Meanwhile, the authors have provided additional experiments to verify the performance.

**Reviewer Scores:**

Reviewer PsdU and PbZX remain positive.

AAag might raise the initial negative score because the authors have made clarification about some details and provide additional experiments.

GNZq didn't understand the proposed algorithm initially and  showed an intention to reconsider the contribution of this paper after the rebuttal.

---

### Decision · Program_Chairs · 2026-01-26

Accept (Poster)